**∂ | Open Peer Review** | Computational Biology | Research Article

# Microbiome time series data reveal predictable patterns of change

Zuzanna Karwowska,[1,2] Paweł Szczerbiak,[1] Tomasz Kosciolek[1,3]

**ABSTRACT**    The human gut microbiome is crucial in health and disease. Longitudinal studies are becoming increasingly important compared to traditional cross-sectional approaches, as precision medicine and individualized interventions are coming to the forefront. Investigating the temporal dynamics of the microbiome is essential for comprehending its function and impact on health. This knowledge has implications for targeted therapeutic strategies, such as personalized diets or probiotic therapy. In this study, we focused on developing and implementing methods specifically designed for analyzing gut microbiome time series. Our statistical framework provides researchers with tools to examine the temporal behavior of the gut microbiome. Key features of our framework include statistical tests for time series properties, predictive modeling, classification of bacterial species based on stability and noise, and clustering analyses to identify groups of bacteria with similar temporal patterns. We analyzed dense amplicon sequencing time series from four generally healthy subjects. Using our developed statistical framework, we analyzed both the overall community dynamics and the behavior of individual bacterial species. We showed six longitudinal regimes within the gut microbiome and discussed their features. Additionally, we explored whether specific bacterial clusters undergo similar fluctuations, suggesting potential functional relationships and interactions within the microbiome. Our development of specialized methods for analyzing human gut microbiome time series significantly enhances the understanding of its dynamic nature and implications for human health. The guidelines and tools provided by our framework support scientists in studying the complex dynamics of the gut microbiome, fostering further research and advancements in microbiome analysis. The gut microbiome is integral to human health, influencing various diseases. Longitudinal studies offer deeper insights into its temporal dynamics compared to cross-sectional approaches. In this study, we developed a statistical framework for analyzing the time series of the human gut microbiome. This framework provides robust tools for examining microbial community dynamics over time. It includes statistical tests for time series properties, predictive modeling, classification of bacterial species based on stability and noise, and clustering analyses. Our approach significantly enhances the methodologies available to researchers, promoting further exploration and innovation in microbiome analysis.

**IMPORTANCE**    This project developed innovative methods to analyze gut microbiome time series data, offering fresh insights into its dynamic nature. Unlike many studies that focus on static snapshots, we found that the healthy gut microbiome is predictably stable over time, with only a small subset of bacteria showing significant changes. By identifying groups of bacteria with diverse temporal behaviors and clusters that change together, we pave the way for more effective probiotic therapies and dietary interventions, addressing the overlooked dynamic aspects of gut microbiome changes.

**KEYWORDS**    gut microbiome, bioinformatics, time series

Address correspondence to Tomasz Kosciolek, tomasz.kosciolek@polsl.pl.

The authors declare no conflict of interest.

See the funding table on p. 22.

The gut microbiome plays a crucial role in human health and disease. The majority of microbiome studies have relied on cross-sectional data, providing only snapshots of its composition at a specific time point (1–3). In recent years, there has been a growing recognition of the importance of longitudinal studies, which enable us to explore the dynamics of the gut microbiome over time (4–10).

To truly understand how a healthy microbiome functions over time, it is essential to study its long-term behavior, including the fluctuation patterns of different bacterial species and the formation of bacterial clusters with similar temporal trends. Such an understanding is pivotal for deciphering the microbiome's influence on health and for devising personalized therapeutic interventions. For instance, insights into the healthy microbiome's temporal dynamics can inform the development of personalized diets, probiotic therapies, and fecal microbiota transplantations, tailoring these interventions to individual needs (11–16).

Focusing on the healthy microbiome's behavior over time allows us to establish a baseline of normal microbial fluctuations and interactions. This baseline is instrumental in identifying deviations associated with disease states, enabling early intervention strategies to restore microbial balance. Moreover, by pinpointing key bacterial species that contribute to a healthy microbiome, we can guide the use of probiotics and other therapies to support microbial health and prevent disease (17). A deeper understanding of the gut microbiome's temporal dynamics is not just about tracking changes but also about leveraging this knowledge to improve health outcomes through precise, personalized interventions. This approach marks a significant shift toward proactive health management, emphasizing the prevention and treatment of conditions linked to the microbiome.

Here, by adopting a rigorous statistical approach, we aim to shed light on the temporal changes in the gut microbiome and unravel its intricate behavior over time. In this study, we investigate the temporal dynamics of the gut microbiome, examining how its composition evolves as a community and how individual bacterial species behave over time. We also explore whether specific clusters of bacteria exhibit similar fluctuations, which could provide insights into potential functional relationships and interactions within the microbiome.

In contrast to the prevailing use of observational approaches in many previous studies (7, 18), our research distinguishes itself by employing statistical methods to analyze human gut microbiome time series data. This distinction is significant as statistical analysis allows for a more rigorous examination of microbial dynamics, enabling the identification of patterns, trends, and associations that may have been overlooked. By applying statistical tests, we not only confirm the consistency of our results with prior findings but also provide a systematic and reproducible framework that quantifies the behaviors of individual bacterial species. This quantitative approach adds depth and reliability to our understanding of the gut microbiome and opens up new avenues for personalized medicine and targeted interventions. The framework is freely available to the community and can be accessed as a GitHub repository at https://github.com/Tomasz-Lab/dynamo.

## RESULTS

In the first part, we describe the behavior of the microbiome over time as a whole. We examine whether it exhibits white noise behavior and is stationary and seasonal, and whether we can predict its change over time. We also demonstrate how the taxonomy changes over time. The second part focuses on the analysis of individual amplicon sequence variants (ASVs) that constitute the microbiome and their behavior over time. We present a methodology through which we describe each ASV using a longitudinal feature vector. Furthermore, we demonstrate the existence of groups of ASVs that exhibit similar behaviors over time. Finally, in the third part, we present the results of graph analysis, which show groups of bacteria that fluctuate together over time.

## Whole community analysis

Since our objective was to analyze long and dense time series to accurately capture the dynamics of the gut microbiome, our in-depth analysis required data sets that met specific criteria. As a result, our literature survey led us to select a relatively small number of data sets. Specifically, we used two publicly available 16S rRNA marker gene sequencing data sets that contain data on the human gut microbiome from four adult individuals with no reported diseases (Table 1). To maintain consistent nomenclature, we refer to each individual using the names originally assigned in the original studies: male and female subjects for the first data set, and donor A and donor B for the second data set. We acknowledge that a sample size of four subjects is relatively small. However, these data sets are the only ones available that meet our specific criteria for this in-depth analysis.

### *The human gut microbiome is individual but stable over time*

Initially, we assessed the temporal dynamics of the human gut microbiome. Principal coordinate analysis (PCoA) analysis revealed distinct clusters for each subject's microbiome, indicating host specificity and dynamic changes over time (Fig. 1A and C; Fig. S1). PCoA plots reveal that each individual's microbial composition is distinct over time, as illustrated in Fig. 1A. Despite the samples being collected across numerous days, their compositions remain remarkably consistent within each individual. However, we observed that time points associated with specific events exhibit deviations from the rest of the data for each subject. For instance, in the female subject (Fig. S1A), the first 50 time points, which displayed lower alpha diversity compared to subsequent samples, were distinctly separated in the PCoA ordination space. Similarly, in donor A (Fig. 1F), time points corresponding to travel events showed a clear divergence from the rest of the time series. Time points following an episode of food poisoning in donor B also demonstrated a notable separation from other samples (Fig. S1B). The male subject's data, however, represented an exception. Not only does the time series vary over time, but there is also a significant division among time points that cannot be attributed solely to temporal factors. However, our understanding of this phenomenon is limited by the absence of comprehensive metadata. To explore the collective dynamics, we computed alpha diversity indices [Shannon's diversity index and Faith's phylogenetic diversity (PD) index] and plotted their fluctuation in time. To assess the temporal dynamics of alpha diversity, a linear regression analysis was conducted, relating alpha diversity indices with time. Despite pronounced fluctuations, the results suggest that alpha diversity tends to oscillate around consistent mean values. Additionally, the model's coefficient for the time variable was close to zero, indicating no significant trend over time and implying relative stability in microbial diversity (Fig. 1B; Fig. S2). For donor A, linear regression analysis demonstrated a trend toward baseline following perturbations (yellow box in the right panel of Fig. 1E) despite day-to-day variations.

Analyzing longitudinal human gut microbiome data from four individuals, we found that while each person has a unique microbiome composition, there are common

**TABLE 1** Data set summary[a]

| Study | Subject ID | Time points | ASV[b] | Additional information |
|---|---|---|---|---|
| Moving pictures of the human microbiome (7) | Male | 443 | 1,253 | Subjects were undergoing antibiotic treatment prior to sampling. |
| | Female | 185 | 551 | |
| Host lifestyle affects human microbiota on daily timescales (8). | Donor A | 365 | 1,524 | Subject was traveling in days 70–122, which caused a change in gut microbiome composition. |
| | Donor B | 252 | 1,569 | Subject suffered from food poisoning between days 150 and 159. |

[a]In this study, we analyzed two dense and extensive 16S gut microbiome time series from four relatively healthy subjects. We detailed the subjects' sex, the number of time points at which each subject was sampled, and the number of unique ASVs in each subject's gut microbiome. Additionally, we provided information on specific factors related to the data set that may influence gut microbiome behavior, such as travel or food poisoning incidents.
[b]ASV, amplicon sequence variant.

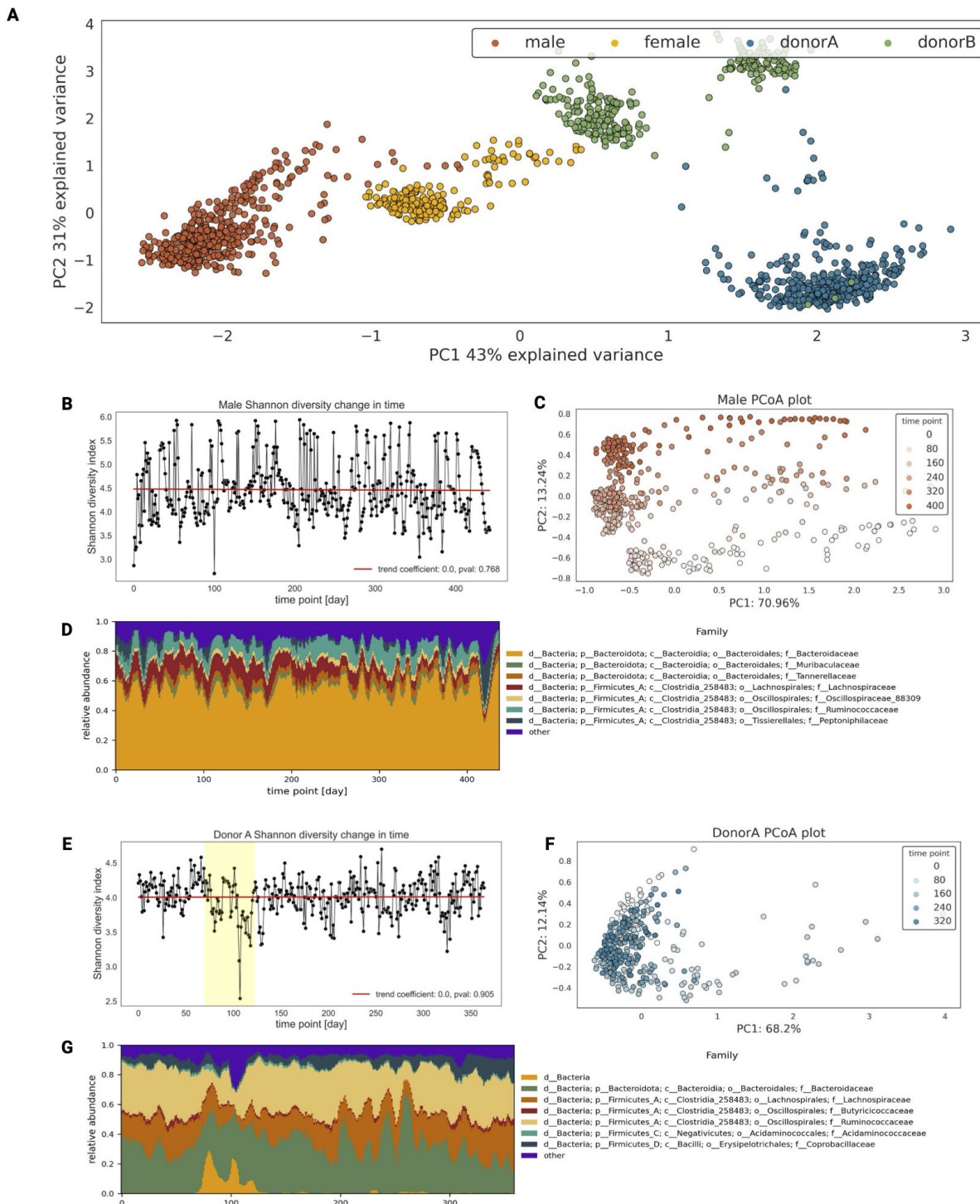

**FIG 1** Analysis of general behavior of gut microbiome in time. (A) The scatterplot of the two first coordinates of principal coordinate analysis (PCA) on Aitchinson distance. (B) Lineplot of male subject in Shannon's diversity index over time. The red line represents the trend in Shannon's diversity index over time, determined by fitting a linear regression model of alpha diversity against time. (C) Scatterplot of the two first coordinates of male subject PCoA on Aitchinson distance matrix. (D) Rolling mean of male subject taxonomy composition on family level in time (window = 14 days). (E) Lineplot of donor A in Shannon's diversity index over time. The red line represents the trend in Shannon's diversity index over time, determined by fitting a linear regression model of alpha diversity against time. The yellow box showcases days 70–122 when the subject was traveling. (F) Scatterplot of the two first coordinates of donor A PCoA on the Aitchinson distance matrix. (G) Rolling mean of donor A subject taxonomy composition on family level in time (window = 14 days).

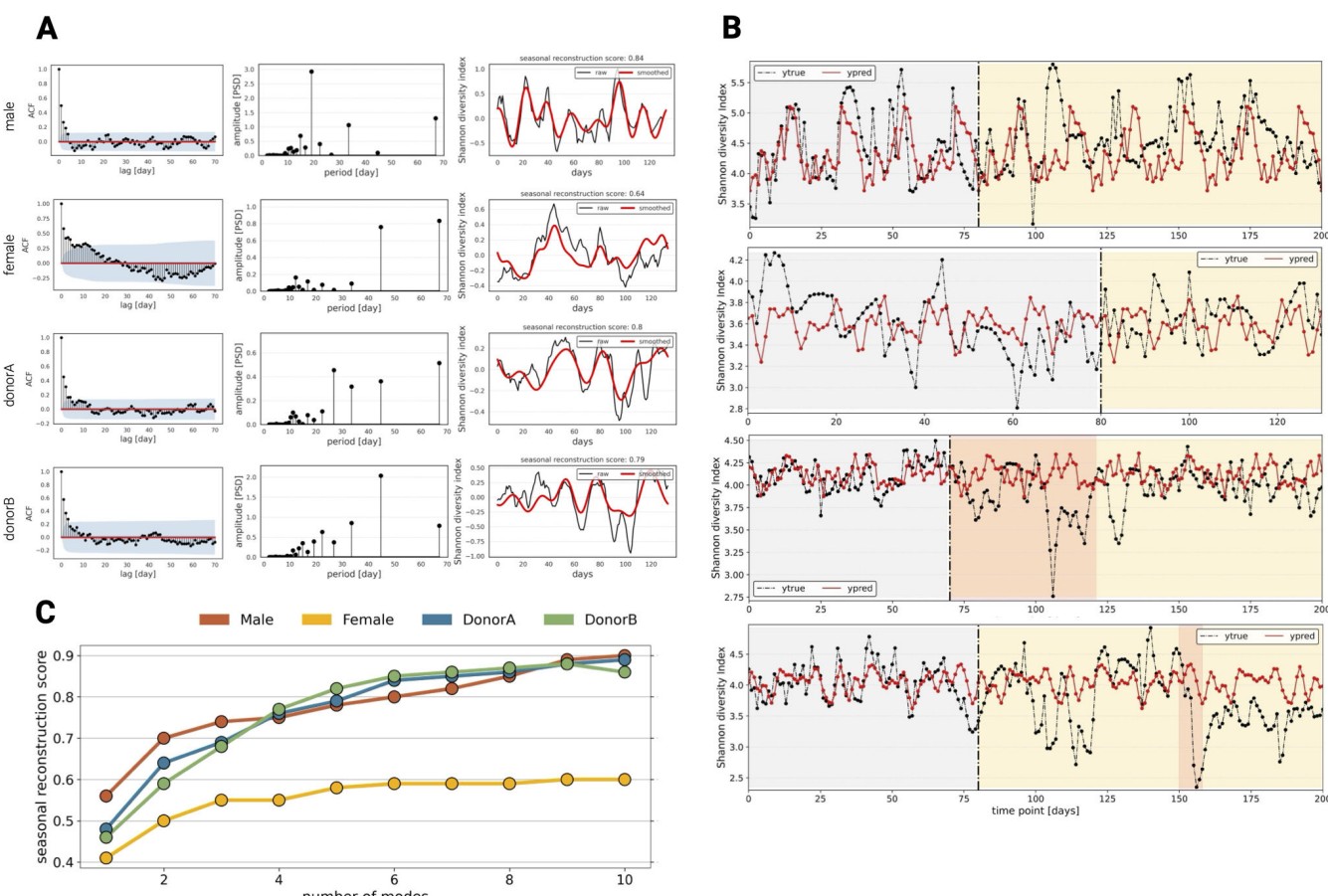

**FIG 2** Shannon's diversity index behavior in time. (A) (Left) Autocorrelation coefficient plots (the blue area indicates the significance level, representing confidence intervals for the autocorrelation coefficient); (middle) spectrograms showing most dominant seasonalities of the human gut microbiome; (right) reconstruction of alpha diversity using five dominant seasonalities plotted against raw alpha diversity change in time. (B) Prediction of alpha diversity change in time using a dynamic AutoRegressive Integrated Moving Average with eXogenous variable (ARIMAX) model with fast Fourier transform seasonalities. The gray area represents the training time point used to train the model, and the yellow area represents the test set. The orange area represents travel and food poisoning, respectively. (C) Plot showing the relationship between a number of used seasonalities to reconstruct alpha diversity and the seasonal reconstruction score.

dominant bacterial taxa (Fig. 1D; Fig. S1). We observed a shared trend where certain bacterial taxa consistently maintained high abundance throughout the duration of the longitudinal data, while others showed lower abundance and temporal variability, appearing intermittently over time (18, 19). Specifically, *Ruminococcaceae*, *Lachnospiracea*, *Bacteroidaceae*, *Oscillospiraceae 88309*, and *Acidaminococcaceae* families dominate in all four subjects. These bacterial families are commonly found in the human gut microbiome and have a shared ability to ferment dietary fibers and produce short-chain fatty acids. They contribute to gut health by providing energy to gut cells and exhibiting metabolic versatility (20–23).

## Predictability of human gut microbiome

In targeted microbiome therapy, the goal is to anticipate the response of the gut microbiome community following the administration of therapy (24). To this end, we aimed to investigate whether the gut microbiome exhibits properties of a predictable time series or if it behaves as a white noise process.

Given the high-dimensional nature of gut microbiome data, the first objective of this study was to investigate the behavior of the human gut microbiome as a unified entity. To achieve this, we employed alpha diversity as a means of biologically reducing the dimensionality of the data to a univariate parameter. Here, we computed two diversity

indices, namely, the Shannon's diversity index and Faith's phylogenetic diversity index, to quantitatively assess the diversity and evolutionary relationships among the microbial taxa present in the human gut (Fig. S2). Finally, we tested each time series for characteristics such as the similarity to the white noise process, stationarity, and the presence of seasonality in the data (Fig. 2).

We investigated whether the human gut microbiome exhibits white noise behavior by analyzing the Shannon diversity index and Faith phylogenetic diversity index of each subject's gut microbiome time series. White noise is a stationary process with consistent statistical properties like mean and variance over time. In white noise, all random variables are independent of each other. This implies that there is no predictable structure or pattern in the sequence of noise values. If alpha diversity displayed white noise behavior, this would imply its variations are random and hard to forecast, making subsequent analysis somewhat redundant. To assess whether alpha diversity aligns with white noise, we investigated autocorrelation, which would imply dependency on preceding values. Furthermore, we evaluated the spectrum flatness score to determine the uniformity of the power across frequencies, a characteristic of white noise, providing insight into the diversity's randomness across different scales. We investigated the presence of unit roots in time series to assess their independence from historical data. A time series is considered stationary if its statistical properties remain constant over time. Conversely, a time series with a unit root is non-stationary, indicating that its mean and variance can vary over time. Our study began by examining autocorrelation using the Ljung–Box test, which assesses the dependency of alpha diversity values on preceding ones. This analysis, alongside spectral flatness scores that indicate uniformity of power across frequencies typical of white noise, provided insights into the randomness of diversity across different scales. We further explored the presence of unit roots in time series to determine their stationarity—a characteristic indicating whether statistical properties remain constant over time using augmented Dickey–Fuller (ADF) and Kwiatkowski–Phillips–Schmidt–Shin (KPSS) tests. The ADF test determines if a process is stationary, while the KPSS test assesses whether a time series is stationary around a trend. The ADF test confirmed that both Shannon's and Faith's phylogenetic indices are stationary across all subjects. The KPSS test showed that the alpha diversities of donor A and donor B are trend stationary. For male and female subjects, the test indicated that these series are trend stationary and might require detrending to become stationary. The Ljung–Box test was run on 70 lags, and we found that the presence of autocorrelation was statistically significant for all 70 lags. The spectral flatness score was calculated for the first 150 days of each time series, aligning with the duration of the shortest time series (150 days) for consistency. The analysis was performed on detrended data. The low flatness scores indicate that all alpha diversities across all subjects do not resemble white noise. According to the definition, a series approaching a flatness score of 1 would exhibit characteristics of white noise. Our findings confirmed that the gut microbiome exhibits both autocorrelations, suggesting a dependence on its previous states. Unit root tests supported the stationary nature of the human microbiome, indicating relatively stable composition over time (Fig. 2A, left panels; Fig. S3; Table 2). Volatility clustering analysis identified regions of increased variance (Fig. S4). However, the lack of metadata limited the understanding of the underlying factors driving this variability. The observed characteristics, including stationarity, autocorrelation presence, and absence of white noise behavior, indicate the predictability of the gut microbiome's future behavior on a general level. However, for a comprehensive understanding, relevant metadata are necessary to address local perturbations. Through the analysis of the autocorrelation function (ACF) and the partial autocorrelation function, we identified the presence of seasonal components—repetitive fluctuations of alpha diversity over time that are not related to specific times of the year—in longitudinal human microbiome data. Our investigation of these repetitive patterns aimed to determine whether a cyclic fluctuation exists within the gut microbiome community.

**TABLE 2** Statistical tests showing alpha diversity behavior in time

| Subject | Alpha diversity | KPSS test $P$ value | ADF test $P$ value | Ljung–Box test $P$ value | Flatness score |
|---|---|---|---|---|---|
| Male | Shannon's diversity index | 0.1 | 0.00[a] | <0.05[a] | 0.04 |
| | Faith's PD | 0.049[a] | 0.00[a] | <0.05[a] | 0.05 |
| Female | Shannon's diversity index | 0.1 | 0.00[a] | <0.05[a] | 0.02 |
| | Faith's PD | 0.092 | 0.00[a] | <0.05[a] | 0.023 |
| Donor A | Shannon's diversity index | 0.1 | 0.00[a] | <0.05[a] | 0.04 |
| | Faith's PD | 0.1 | 0.00[a] | <0.05[a] | 0.14 |
| Donor B | Shannon's diversity index | 0.1 | 0.00[a] | <0.05[a] | 0.03 |
| | Faith's PD | 0.1 | 0.00[a] | <0.05[a] | 0.02 |

[a]Indicates a $P$ value below the significance level of 0.05.

We hypothesize that this fluctuation may be driven by the production of metabolites by one group of bacteria, followed by the subsequent growth of another group. Using fast Fourier transform (FFT) analysis further, we detected multiple dominant seasonal patterns unique to each subject (Fig. 2A, middle panels; Fig. S3C and D). Additionally, low power density spectra indicated the presence of noise and short artificial seasonalities. To assess the reliability of the detected seasonality, we introduced a measure called the seasonal reconstruction score. This score quantifies the Spearman correlation between the raw signal and the signal reconstructed using $N$ Fourier seasonalities. To validate these patterns, we performed inverse fast Fourier transform (IFFT) and calculated the seasonal reconstruction score, finding that at least five seasonalities were required for optimal reconstruction (Fig. 2A, right panels; Fig. S3C and D).

Finally, we aimed to validate the predictability of the human gut microbiome using a dynamic ARIMAX model. The model incorporated dominant seasonal patterns identified through FFT analysis, as standard seasonal autoregressive integrated moving average (SARIMA) models were deemed insufficient. A SARIMA model is a statistical method used to forecast seasonal time series data by incorporating both non-seasonal and seasonal terms, specifically designed to model a single type of seasonality in the data. ARIMAX models were trained on the initial 80 days of data for each subject (70 for donor A, where food poisoning started at day 71), and the parameters were optimized through grid search cross-validation (male $p$: 3, $d$: 0, $q$: 10, $N$: 3; female $p$: 2, $d$: 0, $q$: 7, $N$: 6; donor A $p$: 3, $d$: 0, $q$: 1, $N$: 6; donor B $p$: 4, $d$: 0, $q$: 7, $N$: 6). The model provided a good fit to the training data and satisfactory performance on the test set (Fig. 2B; Fig. S5). However, there were certain regions in the time series where the model was unable to accurately predict fluctuations, as seen in donor A between days 71 and 122, where a drop in alpha diversity occurred due to subject traveling, and in donor B after day 150, during a period of diarrhea (Fig. S5). Nonetheless, our primary aim was to test the self-explanatory nature of the human gut microbiome and evaluate the ability of the model to predict fluctuations in the absence of metadata.

## Individual features analysis

To gain a comprehensive understanding of the behavior of individual bacteria within the human gut microbiome, we generated longitudinal feature vectors for each taxon (represented by ASVs) that captured their characteristics over time. Each feature vector was of length 12 (Table 3). The vectors included general time series characteristics, that is, white noise behavior, stationarity, presence of a seasonal component, and impact on the variability of the overall time series. We simplified the quantification of bacterial behavior by defining two artificial characteristics: noise and seasonal behavior, based on fixed thresholds. Bacteria exhibiting random behavior with no autocorrelation and a flatness score above 0.4 were classified as "noise" (Fig. S6). We used fast Fourier transform analysis to identify dominant seasonal patterns for each taxon. Bacteria were classified as seasonal if their seasonal reconstruction score for five Fourier modes was at least 0.5 (Table 3; see Materials and Methods).

TABLE 3 Characteristics of gut microbiome time series used to construct a feature vector

| General time series characteristics | Mean abundance standard deviation prevalence trend |
| --- | --- |
| White noise behavior | Presence of autocorrelation flatness score |
| Stationarity | ADF test |
| | KPSS test |
| Presence of seasonal component | Dominant seasonality |
| | Seasonal reconstruction score |
| Impact on data variability | First component feature loading |
| | Second component feature loading |

First, we sought to identify unique longitudinal signatures in each subject's gut microbiome by individually analyzing selected longitudinal characteritics (Table 3). The analysis of bacterial abundance in all four subjects revealed distinct patterns. A significant proportion of bacteria were classified as rare or predominantly absent, while a smaller fraction was consistently present in the gut microbiome. Surprisingly, almost half of each individual's microbiome was classified as noise, likely resulting from technical factors and metabolic conditions (25) (Fig. S6 to S8). A considerable portion of ASVs in each subject displayed stationary behavior (Fig. S9). The analysis of seasonality in the human gut microbiome revealed that only a small fraction of bacteria exhibited seasonal behavior, indicating a high variability in their patterns (Fig. S10 to S13). Interestingly, shorter seasonalities were predominantly characterized as noise, while longer seasonalities showed higher seasonal reconstruction scores (Fig. S11). Furthermore, we observe that the seasonal patterns identified in specific bacteria align with the overall seasonality of the gut microbiome composition, as determined through alpha diversity analysis. This consistency underscores the presence of significant seasonal traits in the gut microbiome, suggesting that the observed global seasonality arises from the cumulative seasonal fluctuations of various bacteria (Fig. S12 and S13).

## Longitudinal regimes of the human gut microbiome

After creating a longitudinal characteristics vector for each ASV, we aimed to identify groups of taxa exhibiting similar patterns of behavior in the human gut microbiome. For this purpose, we generated a Spearman correlation matrix from feature vector variables to determine correlated characteristics (Fig. 3).

After analyzing the correlation matrix of longitudinal characteristics, we categorized bacteria into six groups based on their specific longitudinal behavior: (i) prevalent and stable, (ii) prevalent but unstable, (iii) temporal and stable, (iv) temporal but unstable, (v) rare, and (vi) white noise (see Table 4).

The "prevalent and stable" group consists of bacteria that are consistently present in over 90% of the time series and statistical tests confirmed their stationarity. These bacteria show high abundance and are not classified as noise, suggesting their stable presence in the human gut microbiome despite environmental changes. Bacteria classified as "prevalent but unstable" are also present in over 90% of the time series but do not demonstrate stationarity. Due to their high abundance and non-white noise behavior, these bacteria have a significant impact on time series variance (Fig. S14). The "temporal and stable" group comprises bacteria that are present in more than 20% but less than 90% of the time series, exhibit stationarity, and are not classified as noise. Bacteria in the "temporal but unstable" group share the same prevalence patterns as the temporal and stable group but lack stationarity, indicating fluctuating behavior over time. All groups, except for the prevalent and stable group, are classified as a part of the volatile gut microbiome. We hypothesize that these bacteria have more complex metabolic requirements and are more responsive to environmental changes such as diet and medications (19, 26, 27). Additionally, we identified two distinct groups: "rare" bacteria and "white noise" bacteria. Rare bacteria are present in less than 20% of the

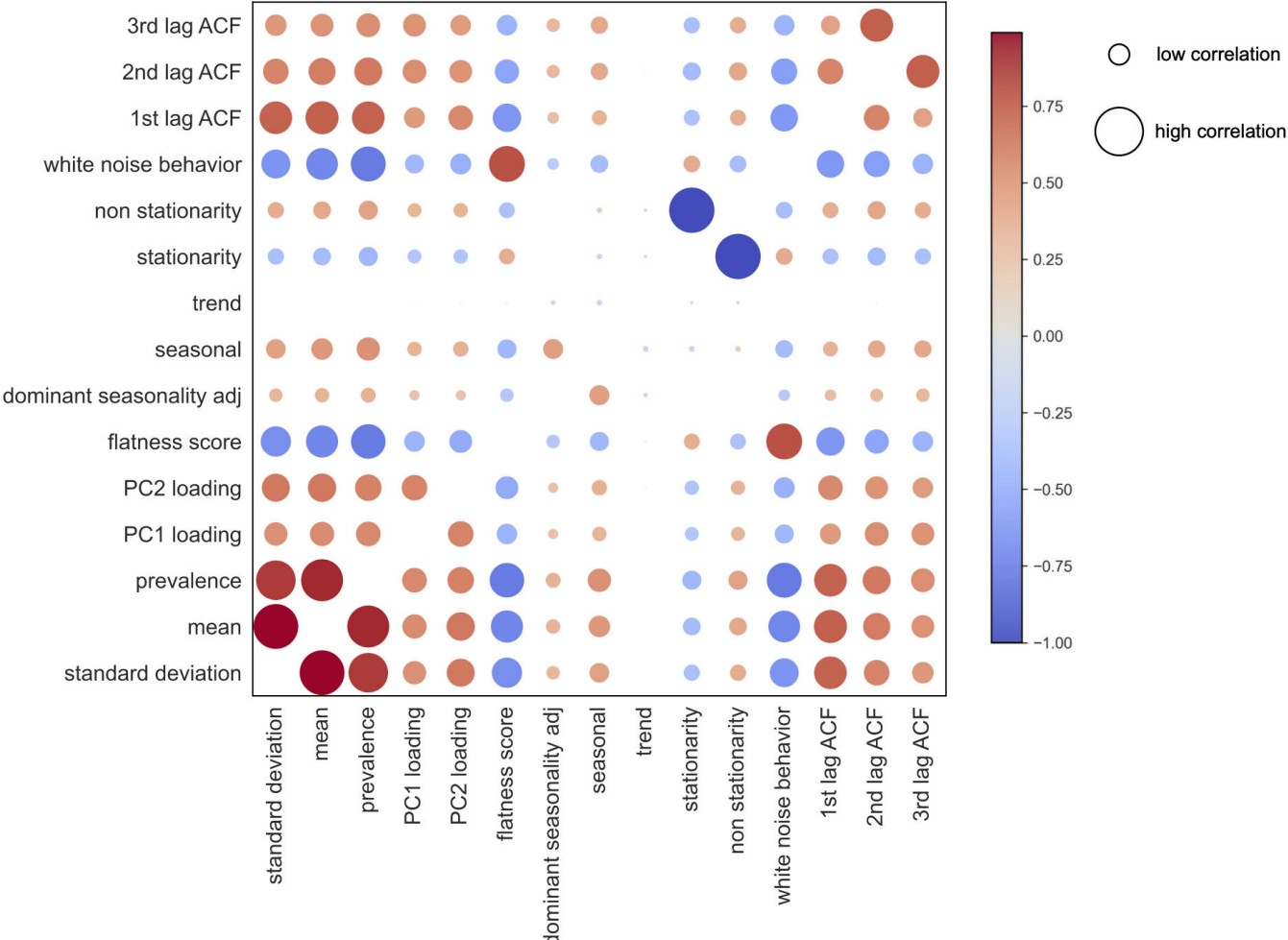

**FIG 3** Correlation of longitudinal characteristics of the human gut microbiome. The Spearman correlation matrix represents the relationships between different longitudinal behaviors of the human gut microbiome. The size of the dots indicates the magnitude of the correlation coefficient, whereas the color denotes whether the correlation is positive (red) or negative (blue).

time series but do not exhibit white noise behavior. We see from further analyses that those bacteria appear in case of specific events such as food poisoning in donor's B time series. On the other hand, white noise bacteria are characterized by low abundance and occurrence. We hypothesize that the white noise bacteria observed in gut microbiome time series data can originate from multiple sources. First, technical noise may be introduced due to variability across sequencing batches, which can lead to the erroneous appearance of some bacteria. This could result from errors in the sequencing

**TABLE 4** Criteria used to define longitudinal regimes of the gut microbiome based on results of statistical tests

| Regime | White noise | | Prevalence | Stationarity | |
| --- | --- | --- | --- | --- | --- |
| | Flatness score | Ljung–Box test *P* value | | ADF *P* value | KPSS *P* value |
| White noise | ≥0.4 | >0.05 | <0.1 | <0.05 | >0.05 |
| Rare | <0.4 | <0.05 | <0.1 | >0.05 | <0.05 |
| Stable prevalent | <0.4 | <0.05 | >0.9 | <0.05 | >0.05 |
| Unstable prevalent | <0.4 | <0.05 | >0.9 | >0.05 | <0.05 |
| Stable temporal | <0.4 | <0.05 | (0.1–0.9) | <0.05 | >0.05 |
| Unstable temporal | < 0.4 | <0.05 | (0.1– 0.9) | >0.05 | <0.05 |

equipment, during the DNA isolation process, or from sample contamination. Second, we propose that biological noise could contribute to the observed fluctuations. This might encompass bacteria exhibiting highly stochastic behaviors that defy prediction. Such unpredictability could stem from metabolic interactions with other bacteria, which are not discernible through 16S rRNA gene sequencing data, or it could be related to the presence of certain opportunistic pathogens in the gut or highly dependent on external factors (such as diet) that we are missing in our data set. Upon thorough analysis of the correlation matrix, it becomes evident that regimes identified through our statistical analyses exhibit distinct characteristics. Features characterized by high prevalence and mean are associated with greater autocorrelation values and a lower flatness score and exert a more substantial influence on the variance of the entire data set, as indicated by higher principal component analysis (PCA) loadings. Conversely, features of lower prevalence demonstrate a minimal impact on time series variance, indicated by lower PCA loadings, exhibit a higher flatness score, and display a lack of autocorrelation.

Next, we performed an analysis to determine the prevalence of each longitudinal regime in every subject. We observed that in all four subjects, bacteria categorized as rare or white noise accounted for over 50% of all rarefied ASVs (Fig. 4A) or even 70%–90% when raw counts are considered (Table S1). The next, most abundant group were the "temporal" bacteria (both "stable" and "unstable"). Moreover, every subject contained a small portion of prevalent and stable bacteria (Fig. 4A). Finally, for all subjects, we analyzed the temporal fluctuations of each longitudinal group over time. Interestingly, although the stable bacteria group representatives were less numerous, they constituted the majority of the time series in terms of abundance. This was followed by a smaller fraction of temporal bacteria (stable and unstable) exhibiting higher volatility. Additionally, rare bacteria appeared for short periods of time, and bacteria defined as white noise, despite being the most numerous group, were nearly undetectable when taking into account their abundance (Fig. 4B).

## Analysis of bacterial clusters

In order to shed more light on bacterial redundancy (i.e., how many bacteria behave similarly) and their relationships, we performed cluster analysis. First, for each subject, we computed proportionality (which is a recommended method for correlation analysis of compositional data (28). The proportionality matrix $\rho$ (of shape $N \times N$, where $N$ is the number of all species in the data set) has been transformed to a pseudosimilarity matrix as $|\rho|$, meaning that any two species that correlate or anti-correlate have a similarity of 1 and 0 if they do not (this reflects the fact that they may be some dependency between them and should be placed close to each other in the graph). Next, we generated NetworkX graphs using spring layout (see details in Materials and Methods). Results are presented in Fig. 5.

We identified three distinct regions in the network: a large cluster in the center of the graph that consists of stochastic and very rare taxa that did not pass the rarefaction step (visible only when both noisy and signal species are considered; see Fig. S15A), medium-size connected components, and a distant cloud of bacteria comprising mostly singletons, i.e., species that do not (anti)co-occur with the others. Figure 5A shows denoised bacteria (i.e., species after rarefaction that do not behave as white noise) colored by the longitudinal regime for $\rho_{thr} = 0.6$, i.e., $|\rho| \geq 0.6$ (see Fig. S15L and M for other thresholds). Clearly, rare and prevalent bacteria cluster out separately and constitute the largest part of the microbiome (in terms of number of species), but we can also notice close connections between practically all regimes (see, e.g., large connected component #1 for male and donor B). We present more examples (colored by abundance, occurrence, taxonomy, PC loading, seasonality, stationarity, and more) and discussion in the supplement. In Fig. 5B and C, we show time evolution of bacteria in different components (cloud, largest connected components, the rest) stratified by the longitudinal regime for male and donor B subjects, respectively (see also Fig. S16A and B for other subjects). First, majority of clouds and largest connected components

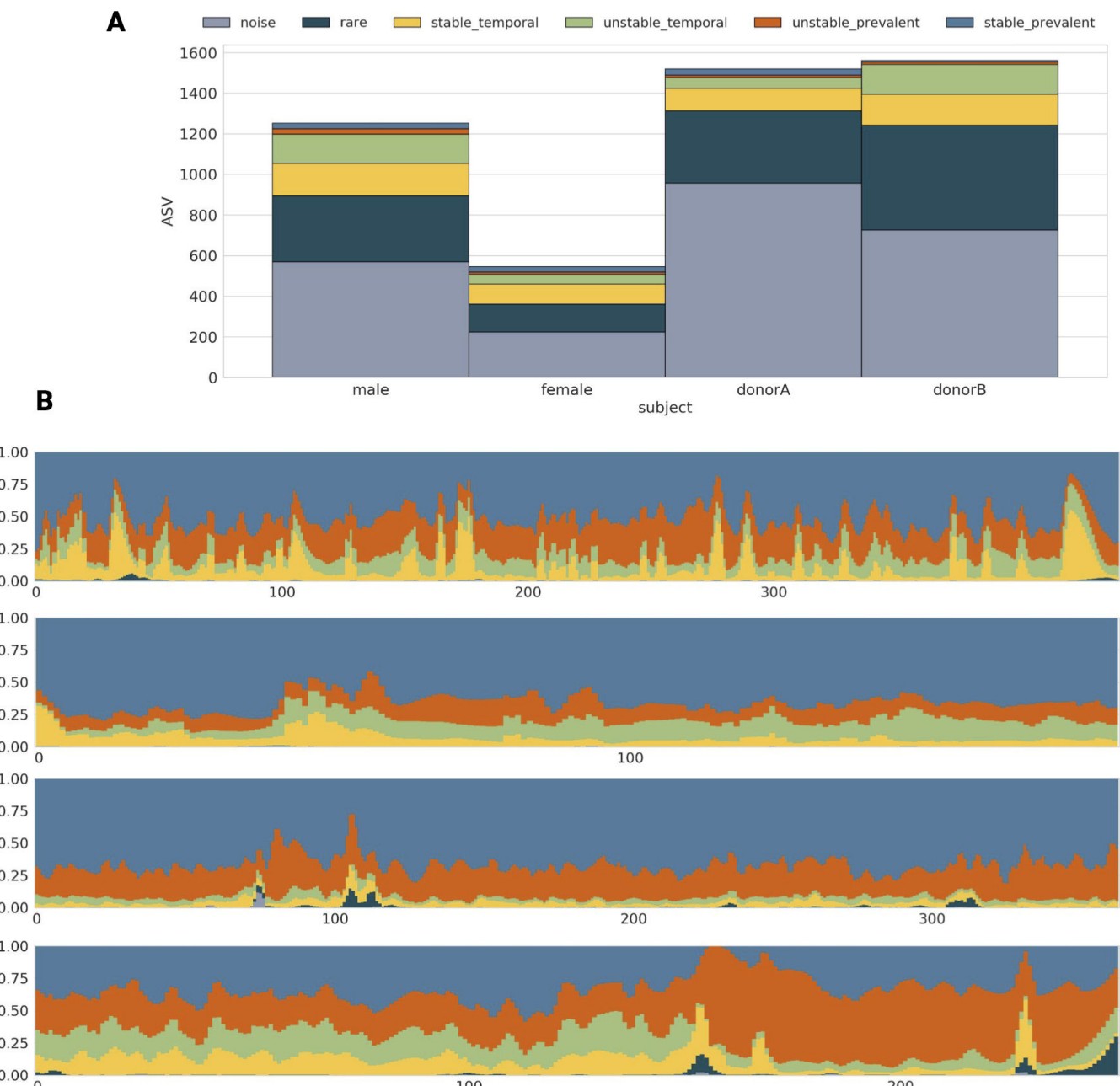

**FIG 4** Longitudinal characteristics of the human gut microbiome. (A) Barplots showing the number of bacteria exhibiting specific regimes in each subject. (B) The fluctuation in counts over time for each longitudinal regime using a stacked barplot. Panels represent male, female, donor A, and donor B regime fluctuations in time. The *x*-axis represents time points in days, and the *y*-axis represents relative abundance of particular regimes.

(except, e.g., the cloud for donor B and connected components for female subject) are dominated (in terms of total counts) by stable species and contain all regimes apart from noisy features that, when included, form a large cluster in the center (Fig. S15A). However, the difference between the cloud and connected components is not obvious and is subject dependent (threshold put on $|\rho|$ also matters but does not change qualitatively the results; see Fig. S16C and D). Second, temporal and unstable species have a large effect on microbiome dynamics, again, irrespective on the data set. In Fig. 5D, we present the PC1 + PC2 loading (the meaningfulness of a given region) of the cloud, the largest connected components, and the rest as a function of $\rho_{thr}$. Clearly, the connected components tend to be more important for explaining the microbiome

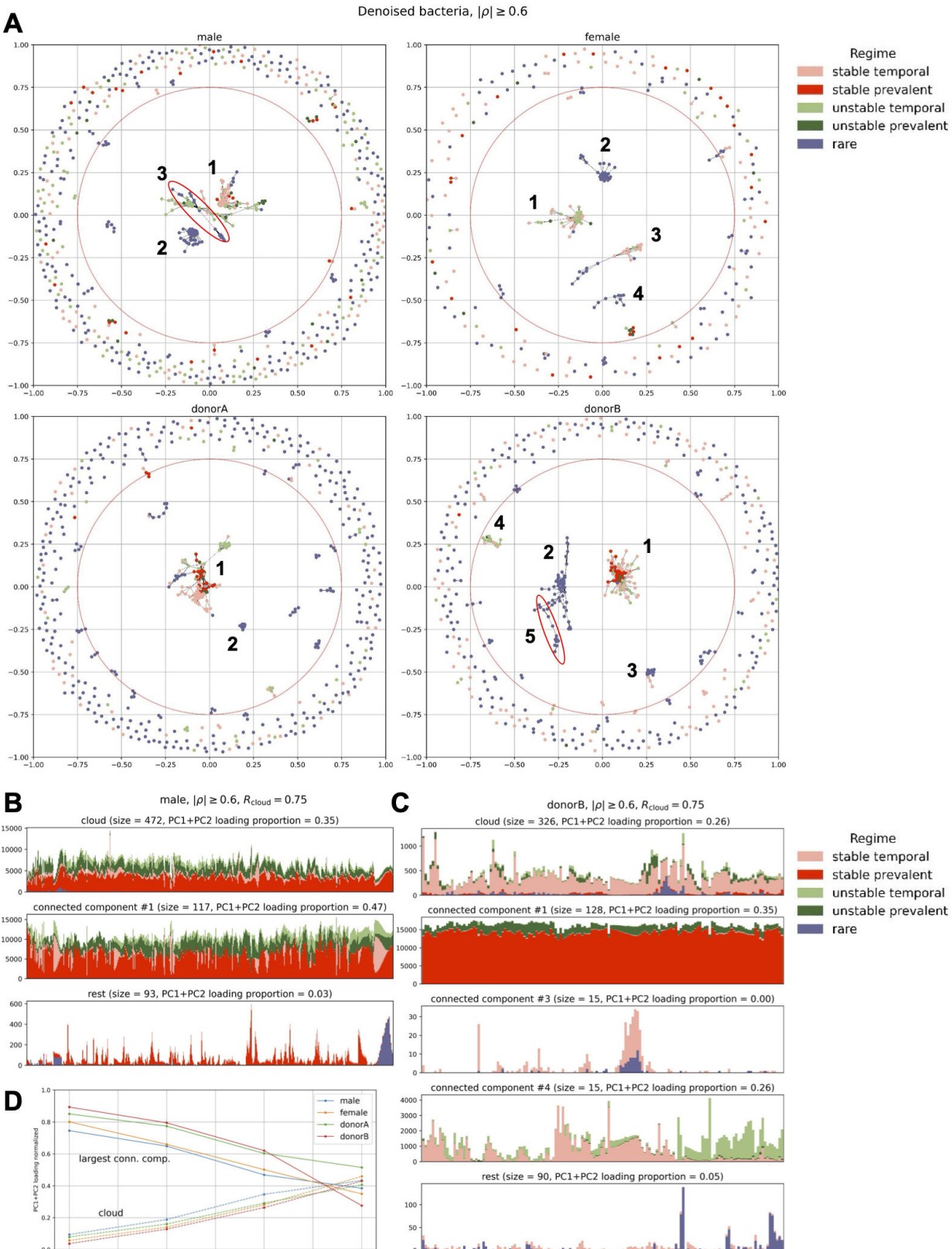

FIG 5 Cluster analysis performed with NetworkX. (A) Network (one panel per subject) of bacterial species (nodes) where connections (edges) represent proportionality equal or stronger than $\rho_{thr} = 0.6$ (equivalent to $|\rho| \geq 0.6$). Colors correspond to longitudinal regimes defined in the Individual Features Analysis section ("none" represents bacteria that did not pass rarefaction). The red circle in each panel separates the inner part from the "cloud." (B) change in total counts

**Fig 5 (Continued)**

after rarefaction over time for male subjects stratified by regime (color) and group of bacteria (panels), i.e., cloud, largest connected components (indicated by numbers in the graphs), and the rest. (C) The same as in panel B but for donor B. In panels B and C, we excluded components comprising only rare species. $R_{cloud}$ represents a diameter that separates the inner part from the cloud. (D) PC1 + PC2 loading against $\rho_{thr}$ for bacteria located in the cloud and largest connected components.

variability with increasing $\rho_{thr}$. However, even more importantly, the size of this effect is subject independent.

The $|\rho|$ threshold for our analyses was selected for its ability to optimally distinguish between different regimes; however, this threshold may differ for other researchers employing our pipeline in various studies. We encourage researchers to experiment with different $|\rho|$ values and explore varying properties of bacterial clusters to determine the most appropriate parameters for their specific needs.

## DISCUSSION

In this study, we leveraged four long and dense time series of the gut microbiome in generally healthy individuals to elucidate its temporal dynamics. Our findings confirm subject-specific microbial signatures (29–32). Through our analysis of alpha diversity trends in the gut microbiome over time, we demonstrate that the gut microbiome behaves as a unified, non-stochastic entity. It exhibits stationarity and predictability based on its previous states, evidenced by the presence of autocorrelation and the efficacy of our predictive model in forecasting its trends. Additionally, our analyses reveal that at the taxonomic level, each individual's gut microbiome is primarily composed of a few dominant groups of bacteria, with occasional temporal blooms of rare bacteria. Despite the small number of analyses of healthy gut microbiome behavior in time, our results are consistent with previous findings that the gut microbiome is host unique and that its composition is stable over time (6, 8, 19, 33–35).

Our study demonstrates the presence of an underlying seasonal pattern in the human gut microbiome. Through the application of the FFT, we identify the existence of multiple dominant seasonalities within the gut microbiome. Moreover, we establish that by utilizing the seasonal component, it becomes possible to predict changes in the human gut microbiome over time with satisfactory performance. The gut microbiome is known to be strongly influenced by external factors, including diet (36). Previous research on the seasonality of the gut microbiome has largely focused on specific data sets from isolated religious groups or indigenous hunter-gatherer communities, whose dietary habits are closely linked to seasonal changes in weather (37–40). We hypothesize that the observed seasonal patterns in the gut microbiome may arise from intricate metabolic interactions among bacteria, which depend on various energy sources derived from the diet and/or their interactions with the external environment. The fluctuations in nutrient availability and dietary composition across seasons could potentially influence the growth and activity of specific bacterial groups, leading to the emergence of distinct seasonal behaviors. Further investigations into the metabolic pathways and interactions within the gut microbiome are warranted to elucidate the underlying mechanisms contributing to these observed seasonal patterns (18). However, without pertinent metadata, annotations of specific bacterial functions (e.g., derived from shotgun metagenomics experiments), or more ubiquitous longitudinal study designs, discerning the precise origins of these seasonal fluctuations remains challenging.

Next, to define how particular bacteria behave in time, we described each bacteria with a longitudinal characteristic vector (Table 3). We create a correlation matrix of longitudinal features to derive groups of features that exhibit a similar behavior. Finally, we define three large groups of bacterial behavior in time: (i) the stable microbiome, (ii) the temporal microbiome, and (iii) noise. We show that the data from the human gut microbiome is, in terms of relative abundance, mostly noise that, we hypothesize, might derive from technical factors. Then, we show that in all four subjects, there exists a small

fraction of bacteria that, despite being few, are highly abundant and are present in more than 90% of the time in the human gut microbiome. Finally, we show that there exists a group of volatile bacteria that, we hypothesize, react more vividly to environmental changes. Our findings align with previous research by Gibbons et al., indicating that the human gut microbiome comprises both predictable autoregressive bacteria and a significant portion of stochastic non-autoregressive bacteria (4, 18). Additionally, our results suggest that diet and other metadata may play a crucial role in gut microbiome dynamics.

Graph analysis (performed with NetworkX) revealed that we can additionally group bacteria in a generally healthy human gut microbiome based on their co-occurrence relationships: noisy bacteria (the largest part in terms of a total number of species) that cluster out together, abundant bacteria (largest part in terms of a total number of counts) that presumably drive microbiome dynamics, and a sizeable part of mostly singletons (denoted as "cloud") that are not (anti)correlated with anything else. The last two regions are highly heterogeneous in terms of their longitudinal regime, which may indicate complex relationships between bacteria from different regimes. The cloud seems to be the most intriguing, especially taxa classified as stable and prevalent that are present in it. However, higher-quality data (larger taxonomic resolution) would be needed to analyze its dynamics and related functions.

Our study aligns closely with previous research, highlighting the coherence of our findings. What distinguishes our approach is the use of rigorous statistical methods, machine learning algorithms, econometric analysis, and graphical tools to examine the behavior of individual bacteria in the human gut. This allows other scientists to more efficiently quantify even large quantities of bacteria and gain new insights into the composition of the human gut microbiome community while analyzing dense time series of human gut microbiome. We believe that our study facilitates scientists in understanding the behavior of bacteria in the human gut and aids in the development of predictive models. Traditionally, researchers have employed a methodology wherein they analyze the top 10% of the most frequently observed bacterial taxa to gain a comprehensive understanding of the microbiome dynamics over time (4, 18, 41). However, our findings demonstrate that while this approach is indeed valuable and the removal of rare bacteria serves as an effective means of reducing dimensionality and noise, it is imperative to acknowledge the presence of bacteria that exhibit temporal patterns, emerging periodically in response to specific conditions within the host's gastrointestinal tract.

Our research underscores the value of employing dense time series analyses in gut microbiome studies and the design of sampling strategies. The demonstrated significance of temporal bacteria suggests that relying solely on single-time-point samples may miss critical taxa, potentially skewing the accuracy of subsequent classification or regression models. We recommend incorporating more frequent temporal series analysis to capture the dynamic nature of the microbiome, offering a clearer resolution of its temporal behaviors. This approach not only enhances the detection of meaningful bacterial activity but also provides a robust framework for more precise and informative conclusions in microbiome research. Thus, we advocate for a methodological shift toward dense time series in microbiome studies to fully leverage the insights such data can provide.

Despite the valuable insights provided by our study, there are certain limitations that should be acknowledged. First, in order to fully understand the dynamics of the microbiome, including seasonality, predictability, and change points, we require additional metadata beyond what was available for the studies we analyzed. This includes factors such as environmental conditions, dietary habits, or health status, and also collecting data following accepted metadata standards (42–44). Second, we aim to build on our findings and predict the entire gut microbiome community using all identified bacteria. Our results indicate that most bacteria exhibit characteristics of a stationary process, suggesting that it should be feasible to forecast their temporal

composition. By advancing this model, we hope to identify the key bacterial drivers that influence changes in the microbiome over time. Finally, to comprehensively annotate the functions of the microbiome, shotgun sequencing data are necessary. While our study focuses on ASVs, it is important to note that sequences and taxonomy alone may be less robust and informative compared to functions. This is because multiple taxa can potentially contribute to the same functions, and different individuals with distinct microbiomes may still exhibit functional similarities (35, 45).

Considering an overwhelming disproportion between cross-sectional and temporal studies, here we demonstrate the utility of microbiome time series data and present a robust and reproducible statistical framework to study it. We show that the gut microbiome changes in a predictable way dictated by individual-specific seasonalities, and that gut bacteria follow one of six longitudinal regimes. However, it is important to note that, at present, there are no available shotgun metagenomic data sets of the healthy gut microbiome, which limits the ability to replicate our analysis. The use of developed methods is not limited to 16S data and can be applied regardless of the sequencing method used. Thus, we believe that with an influx of further investigations incorporating shotgun sequencing data and associated metadata, a more comprehensive understanding of the gut microbiome and its dynamics will emerge. It could be an important step in a transition from observational population-based studies to personalized solutions addressing individual's microbiome composition and its unique dynamics.

## MATERIALS AND METHODS

### Data preparation

#### Data sets

For all analyses in this study, we used two publicly available 16S sequencing gut microbiome data sets containing data of human gut microbiome from four generally healthy adult individuals. Data sets were downloaded from the Qiita repository (https://qiita.ucsd.edu/). Demultiplexing, trimming, and feature table preparation were done using the Qiita framework. Missing time points were interpolated using PCHIP interpolation (see below for details). Interpolated data were rarefied to a 18,000-sequence count threshold. Rarefaction was performed using QIIME 2 (see below for details). Notably, only samples pertaining to the gut microbiome were included in this particular analysis, focusing on the microbial composition specifically within the gut.

#### Data set #1: moving pictures of the human microbiome

The first data set used in this study comprised longitudinal 16S sequencing data obtained from the human microbiota (7) . The data set encompassed two individuals and covered four different body sites. The first individual was a generally healthy adult male who was sampled at four body sites for a duration of 443 days. The second individual, a generally healthy adult woman, was sampled for 185 days.

#### Data set #2: host lifestyle affects human microbiota on daily timescales

The second data set used in this study comprised longitudinal 16S sequencing measurements of the human gut and salivary microbiota dynamics for two generally healthy adult males (8). The data cover a duration of one year, with the first individual sampled for 365 days and the second individual sampled for 252 days.

#### Data preprocessing

For all data sets, raw data underwent preprocessing steps using the Qiita pipeline. These steps included demultiplexing, trimming the sequences to a standardized length of 100 nucleotides, and feature table preparation utilizing the Deblur algorithm. Following the application of the Deblur algorithm, a denoising procedure designed to eliminate

sequencing errors while retaining authentic biological sequences, we acquire a collection of unique, error-corrected sequences. Each sequence delineates an ASV, and together, these sequences constitute the "representative sequences" of the microbial community. The resulting feature tables were then downloaded in Biom format, facilitating subsequent downstream analyses. For phylogenetic analysis, representative sequences were extracted from the data set in FASTA format.

### Interpolation

In all four time series, missing time points were present. These gaps occurred either because the subjects did not provide a fecal sample on those days or the provided samples were not suitable for sequencing. Each time series had missing time points, which are detailed in Fig. S22. Missing time points were interpolated using piecewise cubic hermite interpolation (PCHIP). PCHIP interpolation is a method that approximates a smooth curve or function between data points. It fits a cubic polynomial between adjacent points while ensuring the continuity of the function and its derivative. PCHIP is well suited for microbiome data analysis as it maintains abundances above zero, preserves monotonicity, and avoids overshooting in cases of non-smooth data. Interpolation was performed using SciPy v.1.7.3 Python package with default settings.

### Rarefaction

After interpolation, all four time series underwent rarefaction to 180,000 sequences per sample to mitigate the influence of sequencing depth on alpha diversity analysis. Rarefaction was executed using the QIIME 2 v.2022.2.1 (https://docs. qiime2.org/2022.2.1/), a comprehensive software package designed for microbiome data analysis, and the rarefaction depth threshold was chosen based on rarefaction curves (Fig. S22).

## Whole community analysis

### PCoA between subjects

Aitchison distance between time points among individuals was calculated. Aitchison distance was calculated on non-rarefied data after interpolation. The Aitchison distance is a statistical measure used to quantify compositional differences in relative abundance data, accounting for the constrained nature of compositional space (28). Relative abundance refers to the proportion or percentage of a particular species or taxonomic group within a community, compared to the total number of individuals or groups present. It is a measure of how common or rare a species is relative to others in the same sample. The Aitchison distance is the Euclidean distance between compositions that have been transformed using the centered log-ratio transformation. It possesses desirable properties such as scale invariance, perturbation invariance, permutation invariance, and subcompositional dominance, which are not present in the standard Euclidean distance (46). Aitchison distance matrix was created from calculated distances between the time points of each individual. Next, principal coordinate analysis (PCoA) was used on the distance matrix to reduce data dimensionality. PCoA is a dimensionality reduction technique used to visualize and explore patterns in multivariate data. It converts a distance or dissimilarity matrix into a set of orthogonal axes called principal coordinates, where each axis represents a linear combination of the original variables. Finally, using seaborn v.0.11.2 and Matplotlib v.3.1.3 Python packages, we visualized the first two dimensions of the PCoA results to gain insights into the dissimilarities among the time series.

### PCoA on individual subject

Aitchison distance between time points within the same individual was calculated. Aitchinson distance was calculated on non-rarefied data after interpolation. Next, PCoA was used to reduce the dimensionality of data, and the two first components were visualized using seaborn v.0.12.1 and Matplotlib v.3.5.3 Python packages.

### Phylogenetic tree preparation

Phylogenetic tree construction was performed using QIIME 2 framework pipeline. The pipeline begins by using the mafft program to align representative sequences. Next, the pipeline filters the alignment to remove highly variable positions, which can introduce noise to the phylogenetic tree. Then, FastTree is used to generate the phylogenetic tree from the masked alignment. Finally, midpoint rooting is applied to position the tree's root at the midpoint of the longest tip-to-tip distance in the unrooted tree. For this analysis, we use a rooted tree.

### Alpha diversity calculation

For each of the four individuals, we computed Shannon's diversity index and Faith's diversity index on the rarefied gut microbiome data. Shannon's diversity index (47) measures the richness and evenness of species in a community, focusing on species abundance distribution: $SD = -\sum_{i=1}^{s} p_i \log_{p_i}$, where $s$ is the number of ASVs) and $p_i$ is the proportion of the community represented by $i$th ASV. Faith's phylogenetic index (48) incorporates phylogenetic relatedness among species, emphasizing the evolutionary diversity of the community: $PD_i = \sum_{j \in T} l_{ij}$ branchlen$_j(T)$, where $PD_i$ is Faith's PD for sample $i$; $l_{ij}$ indicates if sample $i$ has any features that descend from node $j$; and branchlen$_j(T)$ indicates the length of the branch to node $j$ in the tree $T$. Faith's phylogenetic index was computed utilizing a rooted tree that was generated following the instructions outlined in the Phylogenetic Tree Preparation section. By constructing the rooted tree using the specified methodology, Faith's phylogenetic index could be accurately calculated and applied to assess the phylogenetic diversity within the studied data set. Alpha diversity indexes were calculated within the QIIME 2 v.2022.2.1 framework.

### Volatility

For each subject's alpha diversity (Shannon's diversity index and Faith's phylogenetic index), the volatility was defined as the average conditional variance observed through-out the entire time series. The conditional variance was obtained by fitting a generalized auto-regressive conditional heteroskedasticity GARCH(1,1) model to the time series, with the model parameters determined through maximum likelihood estimation. A GARCH model is characterized by two primary parameters: p for the order of the autoregressive component (GARCH terms) and q for the order of the moving average component (ARCH terms) that models the conditional variance of the time series. The choice of 1, 1 for the GARCH model parameters was based on the observation that higher values did not yield improved results. The GARCH model was fitted using the arch v.5.3.1 Python package.

### Trend analysis

Trend was calculated using a linear regression model, where time was the explanatory variable and alpha diversity index was the response variable. The explanatory variable was standardized such that it has 0 mean and variance of 1. We defined the trend of the alpha diversity index as a regression coefficient of time. The linear regression model and data scaling were calculated using scikit-learn v.1.0.2 Python package.

### Taxonomy analysis

Taxonomy was assigned to interpolated and rarefied data. To assign taxonomy we first trained a naive Bayes classifier on the GreenGenes2 2022.10 database. Next, we assigned taxonomy to each subject's sequences. Training of the classification as well as taxonomy assignment was performed within QIIME two framework. Plotting was performed using the seaborn v.0.11.2 python Package. For the sake of clarity in visualization, the plot displays only the seven most abundant bacterial families. All other bacterial families have been aggregated under the label "other" to simplify the presentation.

## Autocorrelation and partial autocorrelation

For each subject, the autocorrelation coefficient for 70 lags was calculated on alpha diversity variables (Shannon's diversity index and Faith's phylogenetic index). 95% confidence intervals were used to assess the statistical significance of the autocorrelation coefficient. The selection of 70 lags in gut microbiome data were somewhat arbitrary, as it was primarily chosen to effectively demonstrate the seasonal fluctuations. However, the choice of the specific number of lags can be subjective and dependent on individual preferences and research objectives.

For each subject, partial autocorrelation coefficient for 70 lags was calculated on the alpha diversity variable (Shannon's diversity index and Faith's phylogenetic index). Ninety-five percent confidence intervals were used to assess the statistical significance of the partial autocorrelation coefficient. Both autocorrelation and partial autocorrelation coefficients were calculated using statsmodels v.0.13.5 Python package with default settings and a significance level of 0.05.

## Ljung–Box test

For each subject, the autocorrelation test was run on alpha diversity variables (Shannon's diversity index and Faith's phylogenetic index). The Ljung–Box test (49) is a statistical test used to determine whether a time series is autocorrelated. The lag parameter was initially set to 70, offering a solid default value. Nevertheless, users have the freedom to customize it based on the length of their time series for optimal results. For each lag, we assumed that the autocorrelation is present if the $P$ value is below the significance level of 0.05. The Ljung–Box test was performed using the statsmodels v.0.13.5 Python package with default settings and a significance level of 0.05.

## Unit root tests

In our analysis of the microbiome's overall stationarity, we conducted two unit root tests, namely, the KPSS and ADF, on both alpha diversity measures—the Shannon's diversity index and Faith's phylogenetic index (50). The KPSS test examines whether a time series displays trend or non-stationarity, while the ADF test determines the presence of a unit root, indicating non-stationarity.

In the KPSS test, the null hypothesis asserts that the time series is stationary, with constant statistical properties over time, while the alternative hypothesis suggests non-stationarity, implying variations over time. Conversely, the ADF test's null hypothesis proposes the presence of a unit root in the time series, indicating non-stationarity, while the alternative hypothesis suggests stationarity, implying that the statistical properties of the time series remain constant over time.

Both tests were performed using the statsmodels v.0.13.5 Python package with default settings and a significance level of 0.05.

## Spectrum analysis

For each subject, to detect repetitive patterns in alpha diversity (Shannon's diversity index and Faith's phylogenetic index), we used FFT (51) to detect dominant seasonalities. FFT is an efficient algorithm used to transform a time-domain signal into its frequency-domain representation. It allows for the rapid computation of the discrete Fourier transform by exploiting symmetries and redundancies in the data. To ensure result generalization, we focused on the initial 150 days of each individual's time series. This duration was chosen because it represents the shortest length where no significant events, such as a period of diarrhea affecting the microbiome composition in donor B, occurred. We opted to remove this noisy period to enhance the accuracy of our analysis. First, we removed the trend from the data. To this purpose, linear regression model was fitted as described in the Trend section. Then, the obtained trend was subtracted from the variable. Next, we ran FFT on detrended data. FFT results were plotted using a spectrogram, where we showed on the x-axis the period in units of days, and we showed

on the *y*-axis the period amplitude. All analyses for this part were done using SciPy v.1.7.3 Python package.

### Flatness score

The spectrum flatness score measures the relative balance between the harmonic and non-harmonic components in a signal's frequency spectrum, providing an indication of how "flat" or "noisy" the spectrum is. For each subject's alpha diversity, we calculated flatness score to asses its stochasticity. First, we detrended each time series by fitting a linear regression model to it and subtracting the predicted trend. Then, we calculated flatness score of the detrended time series using the spectrogram analysis. The flatness score was calculated using librosa v.0.10.0 Python package with default settings apart from the n_fft (FFT window size) parameter that was set to the half of the time series length.

### FFT reconstruction

Alpha diversity seasonalities for each subject were sorted based on their amplitude. Starting from the seasonality with the highest amplitude, we employed the IFFT function to reconstruct the alpha diversity by considering only its *N* dominant seasonalities. Subsequently, we calculated the seasonal reconstructions score between the raw alpha diversity index and the alpha diversity index reconstructed using only its dominant seasonality. This analysis was performed for up to 10 dominant seasonalities. To visualize the relationship between the number of seasonalities used for signal reconstruction and the seasonal reconstructions score, we plotted the data using the Python seaborn package. IFFT was calculated with SciPy v.1.7.3 Python package. Spearman correlation coefficient was calculated with statsmodels v.0.13.5 Python package.

### Seasonal reconstruction score

We defined the seasonal reconstruction score as the Spearman correlation coefficient between the raw time series and the time series reconstructed using only its dominant seasonality.

### Alpha diversity prediction

Dynamic Autoregressive Integrated Moving Average (ARIMA) model with a seasonal wave generated using fast Fourier transform was used to forecast the behavior of alpha diversity over time. The training data set consisted of the initial 80 days for each subject. The selection of ARIMA parameters (p, d, and q) and the determination of the number of seasonalities required to create the seasonal wave were performed using a grid search approach. To assess the performance of the models, we utilized the mean average percentage error (MAPE) and Wasserstein distance to measure the similarity between the predicted and true values of alpha diversity. The model with the best performance was chosen for predicting the test data set. For each time series, we predicted the remaining time points and subsequently computed MAPE and Wasserstein distance between the true and predicted values of alpha diversity. To gain insights into the predictability of alpha diversity solely based on the alpha diversity index, we conducted cross-validation by considering consecutive intervals of 20 days. This enabled us to identify periods that were more challenging to predict. By calculating MAPE and Wasserstein distance in this manner, we obtained an evaluation of model performance on the test set. To fit the ARIMA model, statsmodels v.0.13.5 Python package was used. Wasserstein distance was calculated using the SciPy v.1.7.3 Python package. Mean average percentage error was calculated using the scikit-learn v.1.0.2 Python package.

## Individual features

For each ASV, we constructed a longitudinal feature vector describing feature behavior in time. To investigate ASV, we calculated its mean abundance, volatility, prevalence, loading, seasonality, trend, stationarity, and white noise behavior.

### *Mean abundance and standard deviation*

For each ASV, we defined mean abundance as the mean number of reads per day in the whole time series, calculated on interpolated and rarefied data. Mean and standard deviation were calculated with the NumPy v.1.21.6 Python package.

### *Prevalence*

For each ASV, we defined mean prevalence as the percentage of days where ASV is present compared to the whole time series length.

### *Loading*

For each ASV, we defined mean loading as the loading derived from PCoA of data for each subject (see PCoA on Individual Subject). For each subject, we first calculated Aitchinson's distance between all time points. Next, we used PCoA to reduce the dimensionality of data into two components, PC1 and PC2, respectively. Feature loading refers to the cumulative contribution or influence of individual variables (features) on two resulting coordinate axes.

### *Seasonality*

To evaluate the stationarity of each ASV, we initially conducted unit root tests using the KPSS and ADF tests. These tests served to determine whether the ASVs exhibited characteristics of stationarity or non-stationarity. When the KPSS test categorized the ASV as non-stationary, while the ADF test categorized it as stationary, the ASV underwent a detrending process to remove any underlying trend. Conversely, if the KPSS test deemed the ASV as stationary and the ADF test confirmed stationarity, the ASV was differenced by computing the differences between consecutive observations. In cases where both tests indicated non-stationarity, differencing was applied to the ASV. These procedures aimed to enhance the stationarity of the ASV for further analysis and modeling. For each stationary ASV, we found dominant seasonalities using FFT. Using five dominant seasonalities, we used IFFT to create ASV fluctuation in time using only dominant seasonalities. Next, we computed the seasonal reconstruction score between the raw and seasonally reconstructed ASV trajectory. We defined an ASV as seasonal if the correlation coefficient for maximally five seasonalities is above 0.5. We decided on this threshold based on the seasonality analysis shown in Fig. S7. FFT, IFFT, and correlation coefficients were run using the SciPy v.1.7.3 Python package with default settings.

### *Trend*

For each ASV, trend was calculated using a linear regression model where time is the explanatory variable and ASV fluctuation in time is the response variable. The explanatory variable was standardized that it has 0 mean and variance of 1. We defined the trend of the ASV as a regression coefficient of time. The linear regression model data scaling was calculaled using the scikit-learn v.1.0.2 Python package.

### *Stationarity*

To define if a given ASV is stationary, we used two unit root tests (KPSS and ADF; see previous subsection for details). A time series is defined as stationary if both

tests confirm that it does not contain a unit root. Both tests were performed using the statsmodels v.0.13.5 Python package with default settings and a significance level of 0.05.

### White noise behavior

Our subjective definition of white noise behavior was based on two criteria: the absence of autocorrelation and a high flatness score. To assess these criteria, we performed two statistical analyses: (i) the Ljung–Box test statistics was computed for each ASV's 70 lags using the SciPy v.1.7.3 Python package with the default settings and significance level of 0.05; and (ii) the flatness score was calculated for each ASV separately using the librosa v.0.10.0 Python package, also using default settings. To establish a threshold for the flatness score indicating random behavior, we plotted the Ljung–Box test $P$ values against the flatness score. We determined a flatness score threshold of 0.4. Thus, we define a time series as demonstrating white noise behavior if the absence of autocorrelation is validated by the Ljung–Box test, with a $P$ value of >0.05 for all lags, and its flatness score exceeds 0.4. For threshold analysis, see Fig. S6.

### Correlation matrix

We created a Spearman correlation coefficient matrix between longitudinal features using combined data for all ASVs from all four subjects. The Spearman correlation coefficient was calculated using the SciPy v.1.7.3 Python package.

## Cluster analysis

Graphs in Fig. 5 (and in Fig. S15) have been generated using the NetworkX v.2.8.4 Python package (https:// networkx.org/) using spring layout (spring_layout method) with default parameters. Input matrix (so-called pseudosimilarity matrix) has been prepared as follows: (i) for a given data set, raw counts (after interpolation; for donor B, only the first 150 days have been taken into account) were transformed using centered log-ratio transformation with pseudocount equal to 1 using the skbio.stats.composition.clr method (scikit-bio v.0.5.6); (ii) all-vs-all proportionality matrix $\rho$ has been computed as (NumPy v.1.23.5) $\mathrm{rho}(x, y) = 1 - \mathrm{numpy.var}(x - y) / (\mathrm{numpy.var}(x) + \mathrm{numpy.var}(y))$; (iii) for a given $\rho_{\mathrm{thr}}$ (a parameter that controls which species co-occur on anti co-occur), the pseudosimilarity matrix has been constructed as $|\rho|$ with all entries $\leq \rho_{\mathrm{thr}}$ being zero.

In the article, we use $\rho_{\mathrm{thr}} = 0.6$, which provides good separability between different subgraphs (compare with Fig. S15L and M, corresponding to $\rho_{\mathrm{thr}} = 0.5$ and 0.7, where relations between bacteria are more blurred). Certain quantitative results depend on $\rho_{\mathrm{thr}}$, but most of the qualitative conclusions are independent of that parameter (see Discussion in the main text for details).

Hierarchical clustering discussed in the supplement (see Fig. S20 and S21) has been performed using the scipy.cluster.hierarchy Python package (SciPy v.1.10.0). First, linkages were constructed using the linkage method with method="complete" (other values of that argument have been also tested but did not perform that well). Second, clusters were created by cutting the linkage trees using the cut_tree method with hc_height = 2 (again, higher values performed poorly).

## ACKNOWLEDGMENTS

We thank Dr. Ewa Szczurek and Marcin Mozejko from Warsaw University, Poland, for useful discussions on the methodological framework of this work.

This work has been funded by the National Science Centre, Poland (grant 2019/35/D/NZ2/04353).

T.K. and Z.K. conceived the study; Z.K. and P.S. conducted the experiments; Z.K., P.S., and T.K. analyzed the results. All authors reviewed the manuscript. All authors read and approved the final manuscript.

## AUTHOR AFFILIATIONS

¹Malopolska Centre of Biotechnology, Jagiellonian University, Krakow, Poland
²Doctoral School of Exact and Natural Sciences, Jagiellonian University, Krakow, Poland
³Department of Data Science and Engineering, Silesian University of Technology, Gliwice, Poland

## AUTHOR ORCIDs

Zuzanna Karwowska  http://orcid.org/0000-0002-4673-4697
Paweł Szczerbiak  http://orcid.org/0000-0002-7487-6545
Tomasz Kosciolek  http://orcid.org/0000-0002-9915-7387

## FUNDING

| Funder | Grant(s) | Author(s) |
|---|---|---|
| Narodowe Centrum Nauki (NCN) | 2019/35/D/NZ2/04353 | Zuzanna Karwowska |
| | | Paweł Szczerbiak |
| | | Tomasz Kosciolek |

## AUTHOR CONTRIBUTIONS

Zuzanna Karwowska, Conceptualization, Formal analysis, Methodology, Visualization, Writing – original draft, Writing – review and editing | Paweł Szczerbiak, Formal analysis, Methodology, Visualization, Writing – original draft, Writing – review and editing | Tomasz Kosciolek, Conceptualization, Project administration, Resources, Supervision, Writing – original draft, Writing – review and editing

## DATA AVAILABILITY

Code for reproducing figures in this article and many useful methods' implementations used in this work may be found here in https://github.com/Tomasz-Lab/dynamo.

## ADDITIONAL FILES

The following material is available online.

### Supplemental Material

**Supplemental material (Spectrum04109-23-s0001.pdf).** Fig. 1 to 23; Table S1.

### Open Peer Review

**PEER REVIEW HISTORY (review-history.pdf).** An accounting of the reviewer comments and feedback.

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
