## [Reviewer comments · Microbiology Spectrum]

Microbiology Spectrum

Microbiome time series data reveal predictable patterns of change

Zuzanna Karwowska, Pawel Szczerbiak, and Tomasz Kosciolk

Corresponding Author(s): Tomasz Kosciolk, Politechnika Slaska

Review Timeline:

Submission Date:	December 5, 2023
Editorial Decision:	January 13, 2024
Revision Received:	March 15, 2024
Editorial Decision:	March 31, 2024
Revision Received:	May 28, 2024
Editorial Decision:	June 9, 2024
Revision Received:	July 1, 2024
Accepted:	July 5, 2024

Editor: Angela Re

Reviewer(s): The reviewers have opted to remain anonymous.

Transaction Report:

DOI: <https://doi.org/10.1128/spectrum.04109-23>

Re: Spectrum04109-23 (Microbiome time series data reveal predictable patterns of change)

Dear Dr. Tomasz Kosciolk:

Thank you for the privilege of reviewing your work. Below you will find my comments, instructions from the Spectrum editorial office, and the reviewer comments.

Revision Guidelines

Sincerely,
Angela Re
Editor
Microbiology Spectrum

Reviewer #1 (Comments for the Author):

In this manuscript, the authors applied statistical approaches for time series data to analyze dense longitudinal microbiome data at both the community level and individual species level. The temporal patterns of the individual species were closely studied. Overall, the work seems statistically more rigorous than predecessors, and the study is very comprehensive. However, since the major audience of the work are microbiome scientists, who may not have the necessary expertise to understand the concepts from time-series analysis, the authors should explain things in a more intuitive and understandable way. Some illustrations may help the understanding. Another major concern is the validity of applying those time-series methods on microbiome abundance data with severe zero-inflation and non-normality. It is also unknown how to account for the large sampling variability for those

rare species. Presence/absence of these species can be very random due to insufficient sequencing depth. The authors may use some simulations to validate those time-series statistical methods on zero-inflated abundance data. Finally, the categorization of individual species seems ad hoc.

Reviewer #2 (Comments for the Author):

The authors conducted comprehensive statistical analysis to show that human gut microbiome can be predicted based on its previous states. I am impressed by the statistical analysis part and the comprehensive results. However, the authors need to provide more justifications about some of the chosen methods/metrics/data.

Major comments.

It is not clear why the authors choose the four subjects and only four of them. If these are the only available ones, please say that. As there are only four individuals, how can you ensure the conclusions can be generalized to general human gut microbiome? Please discuss these limitations.

Considering that there are many metagenomic data available, please discuss what will be different if using metagenomic data for the study.

As the analysis is not associated with some important events of the hosts (e.g. change of diet, taking medicines etc.), several impacts of the second paragraph of Introduction seem not that relevant to these data and analysis. The authors can make the goal more specific and clear to the readers.

Table 1 needs a more detailed caption.

Page 11: Could the author explain more about the trimming procedure? How did you deal with the rare ASV that was only prevalent once or twice?

Why do you choose 180,000 sequences in Rarefaction?

Page 11: Did the authors compare this distance with the others? such as Bray-Curtis dissimilarities and the phylogeny-aware UniFrac distances.

Editor (Comments for the Author):

1. For phylogenetic analysis, how were representative sequences extracted from the dataset in FASTA format?
2. What do the authors refer to with "consistent mean value" in the comment to Figure 1B? What does the red line represent?
3. Authors are invited to explain and explicitly state the taxonomic rank at which figures (such as Fig. 1D and Fig. 1G) are assembled.
4. By which principle do the authors set out to collect certain data under the "other" label?
5. Could the authors comment on Fig. 1C and similar plots in relation to the variation of bacterial composition along time?
6. Could the authors detail what do the authors mean by relative abundance in Fig. 1G and similar ones?
7. Authors are invited to explicitly state what the colors (gray and yellow) used in Fig. 2B represent
8. Could the authors provide an outlook on technical instruments employable with regard to the caution note in Supplementary Fig. 5 when the authors point out that further investigation and refinement of the model may be necessary to enhance its generalizability and robustness to external variations (assuming metadata are available)?
9. Authors are invited to comment further on white noise definition also by transferring some methodological cautionary notes or figures from the Supplementary Information to the main text.
10. What do the authors mean by missing time points?
11. How do the authors comment on Fig. 3? In the presence of highly correlated features, what actions do the authors undertake? What does the size of the dots represent? Authors are invited to express the informative value of the figure and/or evaluate its transfer to the Supplementary Information. The authors cite this matrix in the discussion. However, the message is unclear.
12. What do the authors mean by writing "we can also notice a higher level of dependency between practically all regimes" in the cluster analysis results?
13. Could the authors deepen the discussion of the possible motivations underlying noise? What technical factors? What about alternative motivations besides technical factors?
14. Could the authors rephrase when, in the discussion, they mention temporal bacteria?
15. The authors are invited to improve the introduction by making it more adherent to the subsequent content.
16. The authors are invited to deepen the points of merit and limits of the analytical pipeline suggested here and to explain how the analysis exemplified by the study cases employed here could be used elsewhere.

17. Why didn't you adopt the same lag for the Ljung-box test as for the correlation computation?
18. What missing time points do the authors impute?

Reviewer #1 (Comments for the Author)

In this manuscript, the authors applied statistical approaches for time series data to analyze dense longitudinal microbiome data at both the community level and individual species level. The temporal patterns of the individual species were closely studied. Overall, the work seems statistically more rigorous than predecessors, and the study is very comprehensive.

However, since the major audience of the work are microbiome scientists, who may not have the necessary expertise to understand the concepts from time-series analysis, the authors should explain things in a more intuitive and understandable way. Some illustrations may help the understanding.

Thank you for the valuable comment. We added a more detailed explanation with examples in the text (Whole community analysis / Predictability of human gut microbiome / page 3).

Another major concern is the validity of applying those time-series methods on microbiome abundance data with severe zero-inflation and non-normality. It is also unknown how to account for the large sampling variability for those rare species. Presence/absence of these species can be very random due to insufficient sequencing depth. The authors may use some simulations to validate those time-series statistical methods on zero-inflated abundance data.

To address the reviewer's concern regarding the validity of applying time-series methods to microbiome abundance data characterized by severe zero-inflation and non-normality, we employed the centered log-ratio (CLR) transformation to mitigate issues related to non-normality. Additionally, we implemented rarefaction techniques to enhance the reliability of our observations, both for detected and undetected species. Diverging from previous studies, we deliberately included even the rarest bacteria in our analysis, categorizing them as highly stochastic in nature. This approach allows us to account for the large sampling variability associated with rare species and addresses concerns about the randomness in the presence/absence of these species due to potentially insufficient sequencing depth. This is now addressed in the manuscript (Methods / Rarefaction and PCoA between subjects / page 13).

Finally, the categorization of individual species seems ad hoc.

We appreciate the reviewer's feedback; however, we must respectfully disagree. Our methodology for categorizing bacteria is grounded in a solid statistical foundation, ensuring that our classification is not only scientifically sound but also thoroughly justified. Our approach distinctively classifies bacteria based on their longitudinal behaviors—such as stationarity, manifestations of white noise, and their effects on time series variance, among others. However, we do agree that the number of different categories and the specific cut-off values for classification may be considered arbitrary and open to refinement in subsequent work.

To enhance clarity and understanding, we have revised the description of our categorization method. These updates are now detailed in the text (Results / Longitudinal Regimes of the Human Gut Microbiome / page 7-9).

Reviewer #2 (Comments for the Author):

The authors conducted comprehensive statistical analysis to show that human gut microbiome can be predicted based on its previous states. I am impressed by the statistical analysis part and the comprehensive results. However, the authors need to provide more justifications about some of the chosen methods/metrics/data.

Major comments.

It is not clear why the authors choose the four subjects and only four of them. If these are the only available ones, please say that. As there are only four individuals, how can you ensure the conclusions can be generalized to the general human gut microbiome? Please discuss these limitations.

We acknowledge that a sample size of four subjects is relatively small; however, our objective was to analyze long and dense time series to more accurately capture the dynamics of the gut microbiome. Unfortunately, these datasets are the only ones available that meet our specific criteria for this in-depth analysis. This issue was acknowledged in the text (Results/Whole community analysis / page 2).

Considering that there are many metagenomic data available, please discuss what will be different if using metagenomic data for the study.

Thank you for the valuable comment. We used all available datasets fulfilling the criteria of dense sampling of mostly healthy individuals. We were unable to locate metagenomic datasets that were both as dense and as extensive as we required, given our focus on analyzing long time series of healthy human gut microbiome. This issue was acknowledged in the text (Discussion / page 12/ lines 303-304).

As the analysis is not associated with some important events of the hosts (e.g. change of diet, taking medicines etc.), several impacts of the second paragraph of Introduction seem not that relevant to these data and analysis. The authors can make the goal more specific and clear to the readers.

Thank you for the valuable comment. Changes have been made as per reviewer's comment (Results/Whole community analysis/The human gut microbiome is individual but stable over time / page 2 -2 / lines 143-144).

Table 1 needs a more detailed caption.

Thank you for the valuable comment. Changes have been made as per reviewer's comment.

Page 11: Could the author explain more about the trimming procedure? How did you deal with the rare ASV that was only prevalent once or twice?

We chose not to exclude very rare bacteria from our analysis, diverging from the common practice of focusing only on the top 15% of prevalent bacteria. This approach, while valid, overlooks the valuable insights rare bacteria can provide, such as those that emerge in response to specific events like food poisoning in donor B. We have only filtered out bacteria during rarefaction, discovering that while many were classified as white noise, a notable subset was categorized as 'rare'. To facilitate broader application of our methodology, we have publicly shared the code used for this analysis (<https://github.com/Tomasz->

Lab/dynamo). This enables other researchers to classify bacteria with our method, which can also serve as a nuanced approach to filtering out noise-contributing bacteria. We added an appropriate note in the text (Discussion / page 11 / lines 297 - 298).

Why do you choose 180,000 sequences in Rarefaction?

We have plotted rarefaction curves and added them in supplementary figure 22. Our choice of rarefaction depth was dictated by our willingness to keep as many samples as possible, while setting the rarefaction threshold as high as possible.

Page 11: Did the authors compare this distance with the others? such as Bray-Curtis dissimilarities and the phylogeny-aware UniFrac distances.

We have calculated Bray Curtis distance and the results can be seen on a PCoA plot (Figure 1 / page 4) and Weighted UniFrac distance can be seen in supplementary Figure 2.

Editor (Comments for the Author):

1. For phylogenetic analysis, how were representative sequences extracted from the dataset in FASTA format?

Thank you for the comment. This issue was acknowledged in the text (Methods / Data preprocessing, page 12).

2. What do the authors refer to with "consistent mean value" in the comment to Figure 1B? What does the red line represent?

The red line illustrates the trend of alpha diversity over time, derived from a linear regression model. In this model, alpha diversity is the dependent variable, and time serves as the independent variable. Specifically, the slope of this line (time coefficient) quantifies the rate of change in alpha diversity over time, indicating how alpha diversity varies with temporal progression.

This issue was acknowledged in the text (Results / The human gut microbiome is individual but stable over time / page 3 / lines 143-144).

3. Authors are invited to explain and explicitly state the taxonomic rank at which figures (such as Fig. 1D and Fig. 1G) are assembled.

This issue was acknowledged in the text (Results / The human gut microbiome is individual but stable over time Fig 1 / page 4).

4. By which principle do the authors set out to collect certain data under the "other" label?

In Figure 1C we plot the progression in time of 7 most abundant bacterial families. All the rest is being merged into the label 'other'. Detailed description has been added in the text (Methods / Taxonomy analysis / page 14).

5. Could the authors comment on Fig. 1C and similar plots in relation to the variation of bacterial composition along time?

Thank you for the comment. This issue was addressed in the text (Results/The human gut microbiome is individual but stable over time / pages 2-3).

6. Could the authors detail what do the authors mean by relative abundance in Fig. 1G and similar ones?

Thank you for the comment. This issue was addressed in the text (Methods / Autocorrelation / page 13 / lines 345-346).

7. Authors are invited to explicitly state what the colors (gray and yellow) used in Fig. 2B represent

Thank you for the comment. The gray section represents the training set and the yellow section represents the test set. Detailed description has been added to figure description.

8. Could the authors provide an outlook on technical instruments employable with regard to the caution note in Supplementary Fig. 5 when the authors point out that further investigation and refinement of the model may be necessary to enhance its generalizability and robustness to external variations (assuming metadata are available)?

Thank you for your insightful comment. We believe that leveraging shotgun metagenomic data could significantly enhance our comprehension of gut microbiome dynamics, allowing for analysis at a higher taxonomic resolution and a deeper understanding of the functional potential. Unfortunately, to the best of our knowledge, there is a lack of extensive and dense shotgun metagenomic datasets on a longitudinal scale, which currently limits the feasibility of our analyses in this direction. Once such data become available, we will be in a position to further refine our model to incorporate causal relationships between bacteria, for example, through metabolic interactions. However, at the moment we cannot suggest any specific solutions as we do not have the data. Nevertheless, we are dynamically working on this subject. We have addressed those limitations in the text (Discussion / page 12 / lines 303-305).

9. Authors are invited to comment further on white noise definition also by transferring some methodological cautionary notes or figures from the Supplementary Information to the main text.

Thank you for your insightful feedback. We have revised Figure 3, incorporating a table that succinctly summarizes the criteria used for defining specific regimes. Additionally, we have enhanced the description of how the correlation matrix facilitated the identification of distinct bacterial behaviors. The presence of high correlations among features within this matrix is intentional and beneficial, as our objective is to determine whether certain characteristics collectively define features.

10. What do the authors mean by missing time points?

The public datasets employed in this analysis comprise samples of the gut microbiome from four generally healthy individuals, with the samples collected on a daily basis. However, not every day is represented in the dataset, owing to either the absence of samples or instances of too low sequencing depth. Samples affected by these issues were removed during the rarefaction step to ensure data quality and consistency.

Thank you for the valuable comment. We added the missing time points plot in supplementary figure 22 and addressed these characteristics of our data in the text (Methods / Data preparation/ Interpolation / page 12).

11. How do the authors comment on Fig. 3? In the presence of highly correlated features, what actions do the authors undertake? What does the size of the dots represent? Authors are invited to express the informative value of the figure and/or evaluate its transfer to the Supplementary Information. The authors cite this matrix in the discussion. However, the message is unclear.

Thank you for your insightful feedback. We have revised Figure 3, incorporating a table that succinctly summarizes the criteria used for defining specific regimes. Additionally, we have enhanced the description of how the correlation matrix facilitated the identification of distinct bacterial behaviors. The presence of high correlations among features within this matrix is intentional and beneficial, as our objective is to determine whether certain characteristics collectively define features (Results / Individual features analysis / Longitudinal regimes of the human gut microbiome / page 7-8).

12. What do the authors mean by writing "c higher level of dependency between practically all regimes" in the cluster analysis results?

Thank you for asking this question. First, it helped us to identify an error in the code responsible for generating the NetworkX graphs - instead of pseudo-similarity matrix we used pseudo-distance matrix. We already corrected the text (including Methods section), all plots (Fig. 5 in the manuscript and all relevant figures in the supplement) and updated the GitHub repository. Fortunately, it didn't change the results qualitatively (this is because we used a relatively high threshold, equal 0.6, put on proportionality), so all conclusions hold. Regarding your question: in connected component #1 for male and donorB subjects we can notice that bacteria belonging to different regimes (rare, stable prevalent, stable temporal, unstable prevalent, unstable temporal) are placed close to each other. This may be caused by the fact that their (anti) co-occurrence patterns are complicated e.g. they (anti) co-occur only in some time regions. To clarify this section, we changed the sentence "higher level of dependency between practically all regimes" into "close connections between practically all regimes" since, indeed, it is hard to estimate the peculiarities (e.g. causal relationship) between bacteria in a given connected component without having additional information like metadata and functions assigned to that taxa. We hope this addresses your concern.

13. Could the authors deepen the discussion of the possible motivations underlying noise? What technical factors? What about alternative motivations besides technical factors?

Thank you for this comment. This issue was addressed in the text (Results/Longitudinal regimes of the human gut microbiome / page 8 / lines 264-266)

14. Could the authors rephrase when, in the discussion, they mention temporal bacteria?

Thank you for your comment. "Temporal bacteria" refers to those categorized within our statistical framework as being prevalent in fewer than 90% of the timepoints for each subject. These bacteria do not exhibit white noise behavior and may be either stationary or non-stationary. We have addressed this distinction in the manuscript (Results/Longitudinal regimes of the human gut microbiome / page 7 / lines 264-265) and have further highlighted the regimes assigned by the authors by italicizing them in the text for clearer differentiation.

15. The authors are invited to improve the introduction by making it more adherent to the subsequent content.
The introduction has been revised in accordance with the suggestions provided by Reviewer 2 and the Editor.

16. The authors are invited to deepen the points of merit and limits of the analytical pipeline suggested here and to explain how the analysis exemplified by the study cases employed here could be used elsewhere
Thank you for the comment. We attempted to improve the take-home message of our work by rewriting the final paragraph of Discussion (page 11-23).

17. Why didn't you adopt the same lag for the Ljung-box test as for the correlation computation?
The lag was unified in both tests to 70 and Methods were updated accordingly. The change of lag did not change the results.

18. What missing time points do the authors impute?

Thank you for the valuable comment. We added the missing time points plot in supplementary figure 22.

Re: Spectrum04109-23R1 (Microbiome time series data reveal predictable patterns of change)

Dear Dr. Tomasz Kosciolk:

Thank you for the privilege of reviewing your work. Below you will find my comments, instructions from the Spectrum editorial office, and the reviewer comments.

- The claim "Here, by adopting a rigorous statistical approach, we aim to shed light on the temporal changes in the gut microbiome and unravel its intricate behavior over time" in the Results paragraph of the abstract does not adhere to the manuscript's content that has a pronounced methodological focus. Authors are invited to conform abstract and introduction to the predominantly technical-analytical message conveyed by the manuscript.
- In the Results paragraph of the abstract "Our study reveals that despite its high volatility, the human gut microbiome is stable in time ..." Please, solve the apparent contradiction.
- The authors are invited to highlight explicitly in the Introduction text that the "computational framework" is made available to the community.
- In "whole community analysis" of the Results section I suggest to invert the order of the sentences like the following: Since our objective was to analyse long and dense time series to accurately capture the dynamics of the gut microbiome, the literature survey according to specific criteria requested by our in-depth analysis led to select a relatively low number of datasets. In particular, we used two publicly available...
- The first instance of "PCoA" occurs without adequate explanation of its full name
- "it's fluctuation in time": please amend the error
- "(yellow box in the right panel of Fig. 1B)": the indication is wrong
- The authors are invited to display the ability of the regression model to explain the data when they draw conclusions on alpha diversity stability in the main text as well as in the supplementary figures.
- "time coefficient if close to 0,": please correct the error
- "the presence of seasonal components in the -repetitive fluctuations of alpha diversity over time that are not related to seasonality or specific times of the year-": Does not a seasonal component relate to seasonality?
- Fig. 2B does not provide sufficient evidences of the ability of the ARIMAX models to predict the dynamics.
- Please explain briefly what SARIMA models. On which statistical basis did the authors claim them insufficient in comparison to the final models?
- In the caption of Suppl. Fig. 5, how can the authors draw the conclusion: "Optimal seasonalities for male, female, donorA, and donorB were found to be 5, 3, 6, and 3, respectively"?
- The comments on Suppl. Figure 8 in the main text does not correspond to the content of the figure. Please, modify Fig. 8 accordingly. Overall, Suppl. Figs. 7-9 and Suppl. Figure 12 are confusing and I suggest creating a single comprehensive figure showing clearly (in relative terms rather than absolute terms) the presence of white noise, stationarity, and seasonality in each individual. This figure can be incorporated in the main text.
- Please clarify the meaning of the sentence "Additionally, the unique features associated with seasonality were found to be consistent with the seasonalities derived from alpha diversity analysis, reinforcing the presence of meaningful seasonal characteristics in the gut microbiome (Supplementary Fig. 12)."
- Supplementary Figure 10 needs substantial reconstruction:
 - o Authors are invited to correct the caption of Suppl. Figure 10 where authors claim: "The x-axis represents the bacterial species, while the y-axis denotes the number of seasonalities required for reconstruction"
 - o The seasonal reconstruction score ranges between 0.2 and 0.5. How can the authors conclude that the model accurately predicts the observed behaviour?
 - o How can the authors state: "The results demonstrate that, in contrast to alpha diversity, different bacterial species exhibit diverse behaviors" and that "Additionally, the analysis highlights that the number of Fourier modes needed to accurately reconstruct the raw signal varies significantly among bacterial species" by commenting on an average index?
- In the section "Longitudinal feature analysis" why don't the authors pinpoint any individual taxon able to predict the dynamic behaviour? This question is the more interesting the clearer is the motivation underlying the claims in Suppl. Fig. 10 where authors say that the predictive mode differs among bacteria.
- "We see from further analyses that those bacteria appear" correct the mistake.
- "proportionality (which is a recommended method for correlation analysis of compositional data)": authors are invited to open-close parentheses
- "we believe that with an influx of further investigations" is redundant with "we expect"
- "Clearly, the cloud tends to be more important for explaining the microbiome variability with increasing pthr" is not supported by Fig. 5D since the loadings tend to be higher for members of the connected components.
- "Aitchison distance" was calculated rather than were calculated
- It's a measure of how common or rare a species:: is relative to others in the same community" is superfluous. Rather, I can't understand from the definition if relative abundance is computed by sample or other entity or if it depends on the analysis set-up.
- "The choice of 1, 1 for the GARCH model parameters": what are these parameters?
- The sub-sections "Autocorrelation" and "Partial autocorrelation" in the Methods section can be grouped.

- Authors need to use the same verbal tense in the Methods section. Now authors use present and past tense intermittently.
- "the number of reads for ASV is present" could become "ASV is present".
- In the cluster analysis, I suggest not citing results obtained without denoising. Otherwise, authors are invited to explain why they deem useful to deliver this information.

In Figure 5 A the numerical labels do not seem to identify the detectable clusters (see donoB and female e.g.).

The authors are invited to explain what they mean by the sentence "the cloud is built from bacteria that are different to anything else and are probably the most intriguing" in the Supplement in relation to the cluster analysis.

- Authors are invited to organize the section "Analysis of bacterial clusters" by emphasizing the opportunities offered by the tool in terms of overlaying the clusters with different features such as abundance, dynamics and so on so forth.
- Discussion should be re-written and reorganized with a focus on the set of analytical procedures that could be applied to dense time series data, on the advantages that could derive and the recommendations to afford more informative conclusions from this type of analysis.
- Please, why do the authors claim that "behaves in a non-stochastic manner as a unified entity," in the Discussion?
- Is it necessary to place the sentences "We analyze the abundance of bacteria exhibiting specific temporal behavior within the gut microbiome of each individual. We create a correlation matrix of longitudinal features to derive groups of features that exhibit similar behavior" in the Discussion?
- What is volatile microbiome mentioned in the Discussion? Please use consistent terminology
- What is "S. Gibbons research"?
- its co-occurrence relationships -> their co-occurrence in Discussion

Revision Guidelines

Sincerely,
Angela Re
Editor
Microbiology Spectrum

Reviewer #1 (Comments for the Author):

The authors have addressed my comments well.

Reviewer #2 (Comments for the Author):

The number of subjects in this study is limited but the authors explained that these are the only data available for this in-depth analysis. When more data become available, the proposed analysis and method can be applied to other subjects.

- The claim "Here, by adopting a rigorous statistical approach, we aim to shed light on the temporal changes in the gut microbiome and unravel its intricate behavior over time" in the Results paragraph of the abstract does not adhere to the manuscript's content that has a pronounced methodological focus. Authors are invited to conform abstract and introduction to the predominantly technical-analytical message conveyed by the manuscript.

Thank you for the valuable comment. Changes have been made as per reviewer's comment (Introduction/page 1/lines 54-56)

- In the Results paragraph of the abstract "Our study reveals that despite its high volatility, the human gut microbiome is stable in time ..." Please, solve the apparent contradiction.

Thank you for the valuable comment. Text has been changed to "Our study reveals that despite its high day-to-day volatility, the long term trend of human gut microbiome is stable in time and can be predicted based solely on its previous states." (Abstract/page 1/lines 34).

- The authors are invited to highlight explicitly in the Introduction text that the "computational framework" is made available to the community.

Thank you for the comment. Changes have been made as per reviewer's comment (Introduction/ page 2/ line 56).

- In "whole community analysis" of the Results section I suggest to invert the order of the sentences like the following: Since our objective was to analyse long and dense time series to accurately capture the dynamics of the gut microbiome, the literature survey according to specific criteria requested by our in-depth analysis led to select a relatively low number of datasets. In particular, we used two publicly available...

Thank you for the comment. Changes have been made as per reviewer's comment (Results/ page 2/ line 64-65).

- The first instance of "PCoA" occurs without adequate explanation of its full name

Thank you for the comment. Changes have been made as per reviewer's comment (Introduction/ page 2/ line 91).

- "it's fluctuation in time": please amend the error

Changes have been made as per reviewer's comment (Introduction/ page 2/ line 92).

- "(yellow box in the right panel of Fig. 1B)": the indication is wrong

Thank you for the comment. Figures were updated as per reviewer's comment (Fig2 and supplementary Figure 5).

- The authors are invited to display the ability of the regression model to explain the data when they draw conclusions on alpha diversity stability in the main text as well as in the supplementary figures.

Thank you for your feedback. We have revised the text to better emphasize the statistical definition of stability within a time series process and have clarified how linear regression models are used to define the stability of alpha diversity over time (Results / The human gut microbiome is individual but stable over time & Predictability of human gut microbiome / page 3)

- "time coefficient if close to 0)": please correct the error

Changes have been made as per reviewer's comment

- "the presence of seasonal components in the -repetitive fluctuations of alpha diversity over time that are not related to seasonality or specific times of the year-": Does not a seasonal component relate to seasonality?

Thank you for the comment. We acknowledge the logical error pointed out and have made the necessary changes in accordance with the reviewer's comments, which can now be found on page 3 of the Results section.

- Fig. 2B does not provide sufficient evidences of the ability of the ARIMAX models to predict the dynamics.

Thank you for the comment. While we appreciate your comments, we must respectfully disagree. Predicting day-to-day fluctuations in the human gut microbiome is an inherently complex challenge. Our objective with this model was to accurately forecast the overarching trends, not necessarily every minor fluctuation. The results, as depicted in our plots, clearly demonstrate that the model successfully captures the stable trends and most of the fluctuations in microbiome behavior over time. We acknowledge that certain periods present greater predictive challenges, primarily due to the absence of detailed metadata, such as daily dietary variations. This is particularly evident in the cases of donorA and donorB, where the model, lacking information on specific events like food poisoning or travel, forecasts general trends rather than precise day-to-day changes in gut microbiome composition. Results showed in Fig 5 show reasonable model's performance. We also acknowledge limitations of our predictive model in the text (Predictability of human gut microbiome / page 5 / line 129).

- Please explain briefly what SARIMA models. On which statistical basis did the authors claim them insufficient in comparison to the final models?

Changes have been made as per reviewer's comment (Results/ page 5/ line 129).

- In the caption of Suppl. Fig. 5, how can the authors draw the conclusion: "Optimal seasonalities for male, female, donorA, and donorB were found to be 5, 3, 6, and 3, respectively"?

Thank you for the comment. We acknowledge the error pointed out and have made the necessary changes in accordance with the reviewer's comments, which can now be found in the Supplementary Fig 5 description and in the main text (Predictability of human gut microbiome / page 5 / line 129).

- The comments on Suppl. Figure 8 in the main text does not correspond to the content of the figure. Please, modify Fig. 8 accordingly. Overall, Suppl. Figs. 7-9 and Suppl. Figure 12 are confusing and I suggest creating a single comprehensive figure showing clearly (in relative terms rather than absolute terms) the presence of white noise, stationarity, and seasonality in each individual. This figure can be incorporated in the main text.

We have rearranged the order of the figures to ensure consistency with the discussions in the main text and enhanced the descriptions to improve comprehensibility. We opted to present the bacteria in absolute

counts rather than relative proportions to more effectively illuminate the variations in bacterial numbers across the subjects.

Additionally, we decided against consolidating all analyses into a single figure, as the current arrangement allows each figure to distinctly showcase different aspects of the temporal characteristics of the human gut microbiome.

- Please clarify the meaning of the sentence "Additionally, the unique features associated with seasonality were found to be consistent with the seasonalities derived from alpha diversity analysis, reinforcing the presence of meaningful seasonal characteristics in the gut microbiome (Supplementary Fig. 12)."

Thank you for the comment. We acknowledge the error pointed out and have made the necessary changes in accordance with the reviewer's comments (Results /page 6).

- Supplementary Figure 10 needs substantial reconstruction:

o Authors are invited to correct the caption of Suppl. Figure 10 where authors claim: "The x-axis represents the bacterial species, while the y-axis denotes the number of seasonalities required for reconstruction"

Changes have been made as per reviewer's comment

o The seasonal reconstruction score ranges between 0.2 and 0.5. How can the authors conclude that the model accurately predicts the observed behaviour?

Thank you for the comment. Thank you for your comment. Supplementary Figure 10A displays a seasonal reconstruction score ranging from 0.2 to 0.8 for some bacteria. Based on this data, we firmly believe that using at least 3-6 different seasonalities will enable us to effectively reconstruct the seasonal behavior of these bacteria.

o How can the authors state: "The results demonstrate that, in contrast to alpha diversity, different bacterial species exhibit diverse behaviors" and that "Additionally, the analysis highlights that the number of Fourier modes needed to accurately reconstruct the raw signal varies significantly among bacterial species" by commenting on an average index?

Thank you for your comment. This sentence summarizes the results shown in both panels of the figure. From panel A, we observe that some bacteria contain more seasonal components than others. To clarify, we have now separated the descriptions of each panel with an overall summary of the figure's findings.

- In the section "Longitudinal feature analysis" why don't the authors pinpoint any individual taxon able to predict the dynamic behaviour? This question is the more interesting the clearer is the motivation underlying the claims in Supp. Fig. 10 where authors say that the predictive mode differs among bacteria.

Thank you for your comment. In this project, our focus was not on using specific bacteria to predict changes in microbiome composition over time; instead, we concentrated on predicting alpha diversity, as it exhibited properties typical of a stationary process. Predicting the entire community composition was outside the scope of this paper. However, we are currently exploring this aspect in a separate project. We have incorporated the reviewers' feedback into the discussion section (pages 11 and 12) and outlined our future research plans accordingly.

- "We see from further analyses that those bacteria appear" correct the mistake.

Changes have been made as per reviewer's comment

- "proportionality (which is a recommended method for correlation analysis of compositional data": authors are invited to open-close parentheses

Changes have been made as per reviewer's comment

- "we believe that with an influx of further investigations" is redundant with "we expect"

Changes have been made as per reviewer's comment

- "Clearly, the cloud tends to be more important for explaining the microbiome variability with increasing pthr" is not supported by Fig. 5D since the loadings tend to be higher for members of the connected components.

Thank you for the comment. We acknowledge the logical error pointed out and have made the necessary changes in accordance with the reviewer's comments, which can now be found in the text (Results / page 9).

- "Aitchison distance" was calculated rather than were calculated

Changes have been made as per reviewer's comment

- "It's a measure of how common or rare a species:: is relative to others in the same community" is superfluous. Rather, I can't understand from the definition if relative abundance is computed by sample or other entity or if it depends on the analysis set-up.

Changes have been made as per reviewer's comment to "It's a measure of how common or rare a species is relative to others in the same sample".

- "The choice of 1, 1 for the GARCH model parameters": what are these parameters?

Changes have been made as per reviewer's comment (Methods / page 13).

- The sub-sections "Autocorrelation" and "Partial autocorrelation" in the Methods section can be grouped.

Changes have been made as per reviewer's comment (Methods / page 13).

- Authors need to use the same verbal tense in the Methods section. Now authors use present and past tense intermittently.

Changes have been made as per reviewer's comment

- "the number of reads for ASV is present" could become "ASV is present".

Changes have been made as per reviewer's comment

- In the cluster analysis, I suggest not citing results obtained without denoising. Otherwise, authors are invited to explain why they deem useful to deliver this information.

Thank you for your comment. Given the classification of bacteria into different regimes and the observation that many bacteria fall within the noise regime, we decided to include noise in our graphs. This addition visually distinguishes between bacteria that do not exhibit stochastic behavior and those that behave like white noise, whose behavior cannot be reliably predicted over time.

In Figure 5 A the numerical labels do not seem to identify the detectable clusters (see donoB and female e.g.).

Changes have been made as per reviewer's comment

The authors are invited to explain what they mean by the sentence "the cloud is built from bacteria that are different to anything else and are probably the most intriguing" in the Supplement in relation to the cluster analysis.

Changes have been made as per reviewer's comment.

- Authors are invited to organize the section "Analysis of bacterial clusters" by emphasizing the opportunities offered by the tool in terms of overlaying the clusters with different features such as abundance, dynamics and so on so forth.

Changes have been made as per reviewer's comment (Results / page 9).

- Discussion should be re-written and reorganized with a focus on the set of analytical procedures that could be applied to dense time series data, on the advantages that could derive and the recommendations to afford more informative conclusions from this type of analysis.

Changes have been made as per reviewer's comment (Discussion / page 11).

- Please, why do the authors claim that "behaves in a non-stochastic manner as a unified entity," in the Discussion?

Changes have been made as per reviewer's comment to "Through our analysis of alpha diversity trends in the gut microbiome over time, we demonstrate that the gut microbiome behaves as a unified, non-stochastic entity. It exhibits stationarity and predictability based on its previous states, evidenced by the presence of autocorrelation and the efficacy of our predictive model in forecasting its trends.". (Discussion / page 9).

- Is it necessary to place the sentences "We analyze the abundance of bacteria exhibiting specific temporal behavior within the gut microbiome of each individual. We create a correlation matrix of longitudinal features to derive groups of features that exhibit similar behavior" in the Discussion?

Changes have been made as per reviewer's comment.

- What is volatile microbiome mentioned in the Discussion? Please use consistent terminology

Changes have been made as per reviewer's comment to "temporal microbiome".

- What is "S. Gibbons research"?

Changes have been made as per reviewer's comment (Discussion / Page 11).

- its co-occurrence relationships -> their co-occurrence in Discussion

Changes have been made as per reviewer's comment.

Re: Spectrum04109-23R2 (Microbiome time series data reveal predictable patterns of change)

Dear Dr. Tomasz Kosciolk:

Thank you for the privilege of reviewing your work and for the care in evaluating previous comments. Below you will find my remaining comments, instructions from the Spectrum editorial office, and the reviewer comments.

- The abstract does not sufficiently emphasise the methodological interest of the article and still includes sentences such as the first sentence in the Results section and the section "Importance of the study" .
- In the section "The human gut microbiome is individual but stable over time" of Results authors claim "(yellow box in the right panel of Fig. 1B)" Fig. 1B does not contain any yellow box.
- The authors do not properly report the decision statistics of a linear regression analysis in the text as regards Fig. 1.
- The authors do not explain what the red line stands for in Fig. 1B and Fig. 1E.
- In Fig. 2 authors state: "blue area shows the significance level". They are invited to specify that the blue area indicate the confidence interval.
- Authors should include the current Supplementary Table 1 in the main text. Supplementary Table 1 does not provide the results for donor B for Shannon diversity index. Moreover, authors are invited to restructure the table by aligning horizontally rather than vertically the results of the tests for the two diversity indices for sake of readability. Moreover, authors are invited to uniform the style by reporting the p-values or the statistical significance in each test and sample. The significance levels for each test should be claimed in the Methods section in the main text. Where are the results on partial auto-correlation?
- Authors state: "Additionally, we examined the presence of a unit root to investigate for non-dependency on historical values, where a unit root indicates no reliance on past data. Unit root tests are employed to determine whether a time series is non-stationary by identifying the presence of a unit root, which indicates that the series will have a time-dependent structure like a changing mean or variance." These sentences are repetitive and conflicting. Authors are invited to shorten the sentences and clarify univocally the interpretation of the presence of unit roots.
- "as well as, Ljung-Box tests" Authors are invited to remove the comma
- Authors should explain the null and alternative hypotheses of the KPSS and ADF tests in the methods clearly. Please, report at least once in the methods the test's name in its extended form.
- Authors claimed "Unit root tests confirmed the stationary nature of the human microbiome, indicating a relatively constant composition over time". Which conclusion is possible to draw from each test? According to the hypotheses under comparison, I guess there exists a preferential order to present the results of the two tests. Finally, what is the possible conclusion that can be extracted overall? Trend-stationarity?
- Lines 165: the observed characteristics? Which ones?
- In Suppl. Fig. 10 don't you think that caption of panel A needs correction? I guess that the x-axis shows the number of modes rather than the bacterial species. Do not bacterial species correspond to dots in the boxplot?
- Authors do not correct the citation of an article "Our results are cohesive with previous S. Gibbons. Gibbons et al. showed"

Revision Guidelines

Sincerely,
Angela Re
Editor
Microbiology Spectrum

Reviewer #1 (Comments for the Author):

The authors have addressed my comments.

- The abstract does not sufficiently emphasise the methodological interest of the article and still includes sentences such as the first sentence in the Results section and the section "Importance of the study" .

Changes have been made as per reviewer's suggestion.

- In the section "The human gut microbiome is individual but stable over time" of Results authors claim "(yellow box in the right panel of Fig. 1B)" Fig. 1B does not contain any yellow box.

Changes have been made as per reviewer's suggestion.

- The authors do not properly report the decision statistics of a linear regression analysis in the text as regards Fig. 1.

Changes have been made as per reviewer's suggestion and regression coefficients have been added to the Figure 1.

- The authors do not explain what the red line stands for in Fig. 1B and Fig. 1E.

Changes have been made as per reviewer's suggestion.

- In Fig. 2 authors state: "blue area shows the significance level". They are invited to specify that the blue area indicate the confidence interval.

Changes have been made as per reviewer's suggestion.

- Authors should include the current Supplementary Table 1 in the main text. Supplementary Table 1 does not provide the results for donor B for Shannon diversity index. Moreover, authors are invited to restructure the table by aligning horizontally rather than vertically the results of the tests for the two diversity indices for sake of readability. Moreover, authors are invited to uniform the style by reporting the p-values or the statistical significance in each test and sample. The significance levels for each test should be claimed in the Methods section in the main text. Where are the results on partial auto-correlation?

Changes have been made as per reviewer's suggestion. Supplementary Table 1 was added to the text as Table 2.

For consistency we have removed partial correlation from both Figure2A and Supplementary Figure 3 as we find that it does not provide additional value to the research shown in the publication.

- Authors state: "Additionally, we examined the presence of a unit root to investigate for non-dependency on historical values, where a unit root indicates no reliance on past data. Unit root tests are employed to determine whether a time series is non-stationary by identifying the presence of a unit root, which indicates that the series will have a time-dependent structure like a

changing mean or variance." These sentences are repetitive and conflicting. Authors are invited to shorten the sentences and clarify univocally the interpretation of the presence of unit roots.

Thank you for the comment. This part (lines 130-131) has been changed to "We investigated the presence of unit roots in time series to assess their independence from historical data. A time series is considered stationary if its statistical properties remain constant over time. Conversely, a time series with a unit root is non-stationary, indicating that its mean and variance can vary over time."

- "as well as, Ljung-Box tests" Authors are invited to remove the comma

Changes have been made as per reviewer's suggestion.

- Authors should explain the null and alternative hypotheses of the KPSS and ADF tests in the methods clearly. Please, report at least once in the methods the test's name in its extended form.

Changes have been made as per reviewer's suggestion (lines 130-131).

- Authors claimed "Unit root tests confirmed the stationary nature of the human microbiome, indicating a relatively constant composition over time". Which conclusion is possible to draw from each test? According to the hypotheses under comparison, I guess there exists a preferential order to present the results of the two tests. Finally, what is the possible conclusion that can be extracted overall? Trend-stationarity?

Changes have been made as per reviewer's suggestion and interpretation of all unit root test was included in the text. (lines 130-131).

- Lines 165: the observed characteristics? Which ones?

Thank you for the comment. This part has been changed to "The observed characteristics, including stationarity, autocorrelation presence, and absence of white noise behavior, indicate the predictability of the gut microbiome's future behavior on a general level. However, for a comprehensive understanding, relevant metadata is necessary to address local perturbations." lines (130-131)

- In Suppl. Fig. 10 don't you think that caption of panel A needs correction? I guess that the x-axis shows the number of modes rather than the bacterial species. Do not bacterial species correspond to dots in the boxplot?

Thank you for the suggestion. The description was changed to "The x-axis represents the number of seasonalities required for reconstruction, while the y-axis depicts the reconstruction score. Each dot corresponds to the score for reconstructing each bacterial species."

- Authors do not correct the citation of an article "Our results are cohesive with previous S. Gibbons. Gibbons et al. showed"

Thank you for the suggestion. The text was changed to “Our findings align with previous research by S. Gibbons, indicating that the human gut microbiome comprises both predictable autoregressive bacteria and a significant portion of stochastic non-autoregressive bacteria. Additionally, our results suggest that diet and other metadata may play a crucial role in gut microbiome dynamics.”

Re: Spectrum04109-23R3 (Microbiome time series data reveal predictable patterns of change)

Dear Dr. Tomasz Kosciolk:

Your manuscript has been accepted, and I am forwarding it to the ASM production staff for publication. Your paper will first be checked to make sure all elements meet the technical requirements. ASM staff will contact you if anything needs to be revised before copyediting and production can begin. Otherwise, you will be notified when your proofs are ready to be viewed.

Sincerely,
Angela Re
Editor
Microbiology Spectrum